# Impact of rheology on probabilistic forecasts of sea ice trajectories: application for search and rescue operations in the Arctic

Matthias Rabatel[1], Pierre Rampal[1], Alberto Carrassi[1], Laurent Bertino[1], and Christopher K. R. T. Jones[2]

[1]Nansen Environmental and Remote Sensing Center, Norway
[2]Department of Mathematics, University of North Carolina, USA

*Correspondence to:* Rabatel (matthias.rabatel@nersc.no or matthiasrabatel@gmail.com)

**Abstract.** We present a sensitivity analysis, and discuss the probabilistic forecast capabilities, of the novel sea ice model *neXtSIM* used in hindcast mode. The study pertains to the response of the model to the uncertainty on winds using probabilistic forecasts of ice trajectories. *neXtSIM* is a continuous Lagrangian numerical model, and uses an elasto-brittle rheology to simulate the ice response to external forces. The sensitivity analysis is based on a Monte Carlo sampling of 12 members. The response of the model to the uncertainties is evaluated in terms of simulated ice drift distances from their initial positions, and from the mean position of the ensemble, over the mid-term forecast horizon of 10 days. The simulated ice drift is decomposed into advective and diffusive parts that are characterised separately both spatially and temporally and compared to what is obtained with a *free-drift* model, that is, when the ice rheology does not play any role on the modelled physics of the ice. The seasonal variability of the model sensitivity is presented, and shows the role of the ice compactness and rheology in the ice drift response at both local and regional scales in the Arctic. Indeed, the ice drift simulated by *neXtSIM* in summer is close to the one obtained with the *free-drift* model, while the more compact and solid ice pack shows a significantly different mechanical and drift behaviour in winter. For the winter period analysed in this study, we also show that, in contrast to the *free-drift* model, *neXtSIM* reproduces the sea ice Lagrangian diffusion regimes as found from observed trajectories. The forecast capability of *neXtSIM* is also evaluated using a large set of real buoy's trajectories, and compared to the capability of the *free-drift* model. We found that *neXtSIM* performs significantly better in simulating sea ice drift, both in terms of forecast error and as a tool to assist search-and-rescue operations, although the sources of uncertainties assumed for the present experiment are not sufficient for a complete coverage of the observed IABP positions.

## 1 Introduction

Large changes in the Arctic sea ice have been observed in recent decades in terms of the ice thickness, extent and drift (e.g. Kwok, 2007; Stroeve et al., 2007; Rampal et al., 2011; Stroeve et al., 2012). These changes, and the underlying driving mechanisms, still need to be fully understood in spite of their being fundamental for building confidence in the forecasting capabilities of current prediction systems. The need for a reliable sea ice prediction platform is particularly felt in the modern context of growing economic opportunities with high societal and environmental impacts. For instance, the dramatic decline of sea ice cover in the Arctic is opening new shipping routes, fishing grounds and tourist destinations as well as access to a significant

portion of the remaining hydrocarbon resources. Associated with this increasing activity are important risks for pollution of the Arctic environment, and risk to human lives. High quality predictions of ocean and sea ice in the polar regions are therefore needed in order to measure the risks, to plan future activities, and to assist operations in real time.

Current short-term (i.e. within 10 days) sea ice forecasting systems integrate either a stand-alone sea ice model (RIOPS (Lemieux et al., 2016; Dupont et al., 2015)), a coupled ice-ocean model (such as, e.g. ACNFS (Hebert et al., 2015), TOPAZ (Sakov et al., 2012) or GIOPS (Smith et al., 2015)), or more seldom a coupled atmosphere-ice-ocean model (GloSea5 (Williams et al., 2015)). Seasonal to decadal climate forecasts are more common and include sea ice as part of the Earth System Models (see, e.g. Carrassi et al. (2016)). The sea ice models used in these systems are usually derived from the work of Hibler III

(1979), and they treat the sea ice as a continuous medium with a viscous-plastic rheology (Hunke and Dukowicz, 1997; Bouillon et al., 2009). In spite of this development, simple free-drift ice (i.e. in the absence of friction and internal forces) forecasts have remained in use by environment agencies (Grumbine, 1998, 2003). The forecast skill of these systems based on a free-drift ice has been evaluated in deterministic mode, when a single "best" forecast is provided: despite the lack of realism in the free-drift assumption, the forecast skill of such systems is seen as difficult to beat (Schweiger and Zhang, 2015).

Probabilistic forecasts, widely used in weather forecasting (Molteni et al., 1996; Leutbecher and Palmer, 2008), are still in their infancy in sea ice forecasting. Probabilistic predictions rely on an ensemble of model simulations (i.e. a Monte Carlo simulation) used to describe the forecast uncertainty stemming from errors in the model parameters, initial and boundary conditions as well as from any external forcing. The resulting cloud of model outputs is used to retrieve statistical information, such as the ensemble mean and its spread (i.e. the standard deviation), that are thus used in place of the deterministic forecast

and to estimate the associated uncertainty, respectively. The multiple simultaneous sources of errors make the forecast accuracy of the ensemble mean usually exceed that of the single deterministic prediction (Leith, 1974; Zhu, 2005), although often the spread underestimates the actual forecast error when the sources of error are not all adequately accounted for (Buizza et al., 2005). Monte Carlo techniques are already common practice in different areas (e.g. Dobney et al., 2000; Hackett et al., 2006;

Breivik and Allen, 2008; Melsom et al., 2012; Motra et al., 2016; Duraisamy and Iaccarino, 2017), and a common tool for sensitivity analysis.

This study concerns the probabilistic forecast capability of the sea ice model *neXtSIM* (Rampal et al., 2016b). The work is carried out by performing a Monte Carlo sensitivity analysis of the model with respect to uncertainties in the surface wind

velocity. The first goal is to highlight the role of the ice rheology on the ice drift: how do the ensemble mean drift and its standard deviation respond to uncertainties in the wind forcing? To answer this question, we compare the ice drift obtained from *neXtSIM* to one obtained from a *free-drift* model. In the second part, we study the skill of the probabilistic forecast using Lagrangian trajectories departing from independent virtual drifting buoys, and compare them with real observations without aiming to make it a key objective. We use the conceptual framework of search and rescue operations where a probabilistic

forecast is commonly used to draw the search area of the ocean where drifting objects are likely to be found (Hackett et al.,

2006; Breivik and Allen, 2008; Melsom et al., 2012). Contrary to these studies, the present simulations are in the context of a "hindcast-forecast", using reanalysed atmospheric forcing fields but assuming that they are affected by errors with the statistical properties that could be expected from a numerical weather forecast in the Arctic. For simplicity, we will use the word "forecast" instead of "hindcast-forecast" throughout the paper.

Our main research tool and object of study is the sea ice model *neXtSIM*. The model *neXtSIM* is based on a Lagrangian numerical scheme and on a continuous approach using a newly developed elasto-brittle ice rheology. This mechanical framework is inspired by the scaling properties of sea ice dynamics revealed by multi-scale statistical analyses of observed sea ice drift and deformation (Marsan et al. (2004), Rampal et al. (2008) and Bouillon and Rampal (2015b)), as well as by the in situ measurements of sea ice internal stresses showing that sea ice deformation is accommodated by Coulombic faulting (Weiss et al. (2007), Weiss and Schulson (2009)). For 40 years, a large variety of sea ice models have been developed. Some, like *neXtSIM*, treat the sea ice as a continuous medium, yet with different rheologies (e.g. Coon et al. (1974) and Hibler III (1979) modelled sea ice as an elasto-plastic material, Hunke and Dukowicz (1997) as an elasto-visco-plastic material, or Dansereau (2016) as an Maxwell-elasto-brittle material), are suitable for high ice concentration ($> 80\%$) while others, that treat the ice as a discrete medium (Hopkins et al., 2004; Wilchinsky et al., 2010; Herman, 2011; Rabatel et al., 2015), are more suitable for low ice concentration ($< 80\%$) such as within the marginal ice zone.

We concentrate here on the impact of the error from the wind field alone. The reasons are twofold: first, the wind is the most influential external force affecting sea ice motion. About 70% of the variance of the sea ice motion in the central Arctic can be explained by the geostrophic winds (Thorndike and Colony, 1982). However, the sea ice response to winds strongly depends on its degree of damage; sea ice responds in a linear way only when it is fragmented into small floes, indeed, in this case, the internal forces are negligible and the inertial term is linearly related to the air and water drags, whereas this behaviour drastically changes when considering a large, continuous and undamaged solid plate. The second reason is due to surface wind velocity fields provided by atmospheric re-analyses contain large uncertainties in the Arctic due to the limited number of observations.

Previous sensitivity analyses of the *neXtSIM* model have been performed with respect to initial conditions and to some key sea ice mechanical parameters (see Sect. 4 in Bouillon and Rampal (2015a)). These analyses consisted in running the model with different values of the input sources. This allowed the authors to explore and quantify the sensitivity of the ice velocity with respect to the ratio between water and air drag coefficients, and of the ice deformation with respect to the compactness parameter value (see Eq. (5)), the sea ice cohesion value (see Eq. (10) in Bouillon and Rampal (2015a)), the initial concentration field, or the initial thickness field. Although these analyses did not use the latest model developments of *neXtSIM* (in particular they did not include the thermodynamics, nor the re-meshing process), the impact of the mechanical parameters on the ice deformation can still be considered as valid.

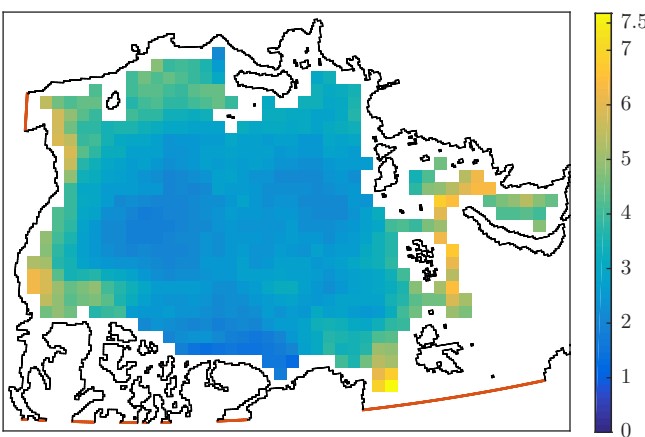

**Figure 1.** Systematic errors in the neXtSIM ice velocities, compared to observations from the OSI-SAF dataset. Simulated and observed ice drift are averaged over the period from 1 January 2008 to 30 April 2008. The cells with less than 28 observations over the winter are masked. The colour scale represents the velocity in $km\ day^{-1}$.

Systematic errors in the mean sea ice drift are evaluatd by averaging modelled and observed drift from the OSI-SAF dataset (Lavergne and Eastwood, 2015) over the period between 1 January 2008 and 30 April 2008 and over boxes of $100 \times 100\ km^2$ covering the whole Arctic (see Fig. 1). The largest differences between the observed and simulated mean ice drift are located in the Beaufort, Chukchi, Kara and Barents Seas and Fram Strait and in some areas of the East Siberian sea. In the rest of the domain the error on the mean winter drift is only less than 3 $km\ day^{-1}$, consistently with Rampal et al. (2016b).

This paper is organized as follows: Sect. 2 gives a general presentation of the sea ice model *neXtSIM* with the main equations describing the sea ice dynamical behaviour; Sect. 3 presents the details of the sensitivity analysis based on a Monte Carlo sampling, including the description of the quantities of interest, the construction of the wind perturbations, and the general experimental setup. In the same Sect. 3, we also define the *free-drift* model that will be used for comparison and benchmark against *neXtSIM*. Section 4 discusses the results for the ensemble mean, spread and the evaluation of the forecast skills comparing *neXtSIM* to the *free-drift* model. Final conclusions are drawn in Sect. 5.

## 2  General information on the model *neXtSIM*

In this section, we provide a general description of *neXtSIM*. Deliberately, we choose to not go through all model equations here, but rather list those that are needed to get an overall understanding of how the model works, and that are relevant for the present study. For a more detailed description of the model see Bouillon and Rampal (2015a) and Rampal et al. (2016b).

*neXtSIM* is a continuous dynamic-thermodynamic sea ice model. It uses a pure Lagrangian advection scheme, meaning that the nodes of the model mesh are moving at each time step according to the simulated ice motion. The model mesh is therefore changing over time, it is not spatially homogeneous, and it can locally become highly distorted, that is, when and where the ice motion field is showing strong spatial gradients. In this case, a local and conservative re-meshing procedure is applied in order to keep the numerical integrity of the model and the spatial resolution of the grid approximatively constant during the simulation. The equations are discretised on a triangular mesh and solved using the classical finite element method, with scalar and tensorial variables defined at the center of the mesh elements, and vectors defined at the vertices. The model is using a mechanical framework that has been developed recently (Girard et al. (2009) and Bouillon and Rampal (2015a)), and which is based on the Elasto-Brittle (EB) rheology. The brittle mechanical behaviour of the sea ice is simulated by calculating the local level of damage in each grid cell, a variable which is not considered in classical viscous-plastic sea ice models typically used in the sea ice modelling community. Sea ice thermodynamic, which is parametrised in *neXtSIM* as in the zero-layer model of Semtner (1976), controls the amount of ice formed or melted at each time step. When a volume of new (and therefore undamaged) ice is formed within a grid cell by thermodynamical freezing, the mechanical strength of the total volume of ice covering that cell is partially restored, and the new damage value is computed as a volume-weighted mean. Note however that the damaging process is very fast (i.e. about few minutes) while the mechanical healing process is occurring over much slower time scales of about several weeks. The sea ice variables used in *neXtSIM* are the following: $h$ and $h_s$ are the effective sea ice and snow thickness respectively (ice and snow volumes per unit area); $A$ is the sea ice concentration (bounded to 1); $d$ is the sea ice damage ranging from 0 (undamaged ice) to 1 (fully damaged); $\boldsymbol{u}$ is the horizontal sea ice velocity vector; and $\boldsymbol{\sigma}$ is the ice internal stress tensor. The model has two ice thickness categories: ice and open water.

The evolution equations for $h$, $h_s$ and $A$ (here denoted $\phi$) have the following generic form

$$\frac{D\phi}{Dt} = -\phi \nabla \cdot \boldsymbol{u} + S_\phi, \tag{1}$$

where $\frac{D\phi}{Dt}$ is the material derivative of $\phi$, $\nabla \cdot \boldsymbol{u}$ the divergence of the horizontal velocity and $S_\phi$ a thermodynamical sink/source term. The evolution of sea ice velocity comes from the following sea ice momentum equation, integrated over the vertical,

$$m\frac{D\boldsymbol{u}}{Dt} = \nabla \cdot (\boldsymbol{\sigma} h) - \nabla P + \boldsymbol{\tau}_a + \boldsymbol{\tau}_w + \boldsymbol{\tau}_b - mf\boldsymbol{k} \times \boldsymbol{u} - m\boldsymbol{g}\nabla\eta, \tag{2}$$

where $m$ is the inertial mass, $P$ is a pressure term, $\boldsymbol{\tau}_a$ is the surface wind (air) stress, $\boldsymbol{\tau}_w$ is the ocean (water) stress and $\boldsymbol{\tau}_b$ is the basal stress in case of grounded ice parametrised as in Lemieux et al. (2015). The last terms are the Coriolis parameter, $f$, the upward pointing unit vector, $\boldsymbol{k}$, the gravity acceleration, $\boldsymbol{g}$, and the ocean surface elevation, $\eta$. The internal stress $\boldsymbol{\sigma}$ is computed as in Bouillon and Rampal (2015a) and Rampal et al. (2016b). Its evolution equation can be written as

$$\frac{D\boldsymbol{\sigma}}{Dt} = \frac{\Delta d}{Dt}\frac{\partial \boldsymbol{C}}{\partial d} : \boldsymbol{\epsilon} + \boldsymbol{C}(A, d) : \dot{\boldsymbol{\epsilon}}, \tag{3}$$

where $d$ is the damage and $\dot{\boldsymbol{\epsilon}}$ is the deformation rate tensor defined as $\dot{\boldsymbol{\epsilon}} = \frac{1}{2}\left(\nabla\boldsymbol{u} + (\nabla\boldsymbol{u})^{\mathrm{T}}\right)$. $\boldsymbol{C}$ can be written as

$$\boldsymbol{C} = \frac{E(A, d)}{(1-\nu^2)}\begin{bmatrix} 1 & \nu & 0 \\ \nu & 1 & 0 \\ 0 & 0 & \frac{1-\nu}{2} \end{bmatrix} \tag{4}$$

with $\nu$ being the Poisson's ratio while $E(A, d)$ the effective elastic stiffness of the ice which depends on the ice concentration $A$ and the damage $d$ according to

$$E = Y e^{-\alpha(1-A)}(1-d) \tag{5}$$

where $Y$ is the sea ice elastic modulus (Young's modulus) and $\alpha$ is the so-called compactness parameter.

The evolution equation for the damage is written as:

$$\frac{Dd}{Dt} = \frac{\Delta d}{\Delta t} + S_d, \tag{6}$$

where $\Delta d$ is a damage source term calculated as in Rampal et al. (2016b) (Eq. (8)), and $S_d$ is thermodynamical sink term which depends on the volume of new and undamaged ice formed over one time step as well as on time (See Rampal et al. (2016b), Sect. 2.3 for more details).

The air and oceanic drags, respectively $\boldsymbol{\tau}_\mathrm{a}$ and $\boldsymbol{\tau}_\mathrm{w}$ in Eq. (2), are written as a force per unit area in the quadratic form using the associated turning angle (Leppäranta, 2011)

$$
\begin{aligned}
\boldsymbol{\tau}_\mathrm{a} &= \rho_a C_a \left\| \boldsymbol{u}_\mathrm{a} - \boldsymbol{u} \right\| R_{\theta_a} \left( \boldsymbol{u}_\mathrm{a} - \boldsymbol{u} \right) \\
\boldsymbol{\tau}_\mathrm{w} &= \rho_w C_w \left\| \boldsymbol{u}_\mathrm{w} - \boldsymbol{u} \right\| R_{\theta_w} \left( \boldsymbol{u}_\mathrm{w} - \boldsymbol{u} \right)
\end{aligned}
\tag{7}
$$

where $\|.\|$, $R_{\theta_a}$, $R_{\theta_w}$, $\boldsymbol{u}_\mathrm{a}$, $\boldsymbol{u}_\mathrm{w}$, $\rho_a$, $\rho_w$, $C_a$ and $C_w$ are, respectively, the Euclidean norm in $\mathbb{R}^2$, the rotation matrix through the angle $\theta_a$ and $\theta_w$, the wind velocity, the ocean current, the air density, the water density, the air drag coefficient and the

water drag coefficient. The values of the model parameters that are used for the simulations presented in this paper are listed in Table 1.

## 3   Sensitivity analysis

### 3.1   Methodology

In this study, we perform a sensitivity analysis using a statistical approach based on Monte Carlo sampling of the model in-

puts. We focus on the response of the model to the uncertainties in the wind velocity field. In particular, we are looking at the response of sea ice drift to wind perturbations representing these uncertainties. Our methodology is based on simulating Lagrangian trajectories of virtual buoys using an ensemble run of the *neXtSIM* model forced by slightly different (i.e. perturbed) wind forcing (see Sect. 3.2 for more details on the generation of the perturbed winds).

The velocity of a given virtual buoy is calculated on-line, at each time step, as a linear interpolation of the velocities simulated at the nodes of the mesh element containing that buoy (see Lagrangian approach in Sect. 2). Each virtual buoy is associated with an initial position $\boldsymbol{x}_0 \in D$, with $D$ being the initial domain, and a start date $t_0 \in Y$ where $Y$ is the time period of interest of this study (see Sect. 3.2 for more details). A buoy trajectory is denoted $\boldsymbol{g}(\boldsymbol{x}_0, t_0, t)$ with $t \in [t_0, T]$, and where $T$ defines the duration of the individual simulations. For each initial position $\boldsymbol{x}_0$ and start date $t_0$, we simulate $N$ trajectories $\{\boldsymbol{g}_i\}_{i \in \{1,...,N\}}$

**Table 1.** Parameters used in the model with their values for the simulations performed for this study.

| Symbol | Meaning | Value | Unit |
|---|---|---|---|
| $\rho_{\mathrm{a}}$ | air density | 1.3 | $\mathrm{kg\,m^{-3}}$ |
| $c_{\mathrm{a}}$ | air drag coefficient | $5.1 \times 10^{-3}$ | – |
| $c_{\mathrm{a}}$ | air drag coefficient (for FD) | $3.2 \times 10^{-3}$ | – |
| $\theta_{\mathrm{a}}$ | air turning angle | 0 | ° |
| $\rho_{\mathrm{w}}$ | water density | 1025 | $\mathrm{kg\,m^{-3}}$ |
| $c_{\mathrm{w}}$ | water drag coefficient | $5.5 \times 10^{-3}$ | – |
| $\theta_{\mathrm{w}}$ | water turning angle | 25 | ° |
| $\rho_{\mathrm{i}}$ | ice density | 917 | $\mathrm{kg\,m^{-3}}$ |
| $\rho_{\mathrm{s}}$ | snow density | 330 | $\mathrm{kg\,m^{-3}}$ |
| $\nu$ | Poisson coefficient | 0.3 | – |
| $\mu$ | internal friction coefficient | 0.7 | – |
| $Y$ | elastic modulus | 9 | GPa |
| $\Delta x$ | mean resolution of the mesh | 10 | km |
| $\Delta t$ | time step | 200 | s |
| $T_{\mathrm{d}}$ | damage relaxation time | 28 | days |
| $c$ | cohesion parameter | 8 | kPa |
| $\alpha$ | compactness parameter | $-20$ | – |

from $N$ model runs, each one corresponding to a different realisation of the wind forcing. If a buoy ends up in an ice-free element, it is then untracked further and its trajectory discarded from the remaining analysis.

For each ensemble member (trajectory), we define the following Euclidean distances $\forall i \in \{1, \dots, N\}$,

$$r_i(t) = \|\boldsymbol{g}_i(\boldsymbol{x}_0, t_0, t) - \boldsymbol{x}_0\|$$
$$b_i(t) = \|\boldsymbol{g}_i(\boldsymbol{x}_0, t_0, t) - \boldsymbol{B}(t)\|,$$

where the quantity $r_i(t)$ is the distance of the member position at time $t$, $\boldsymbol{g}_i(\boldsymbol{x}_0, t_0, t)$, from its departure origin, $\boldsymbol{x}_0 = \boldsymbol{g}_i(t = t_0)$. The second quantity, $b_i(t)$, represents the distance between the member position at time $t$ and the ensemble mean position (i.e. the barycentre, $\boldsymbol{B}(t)$, of the ensemble), $\boldsymbol{B}(t) = 1/N \sum_{i=1}^{N} \boldsymbol{g}_i(\boldsymbol{x}_0, t_0, t)$, at the same time $t$ (see the top panel of Fig. 2). We make use here of the convention of using boldface for vectors and matrices and normal face for scalar quantities; hereafter, we drop the explicit mention of the dependence on $\boldsymbol{x}_0$ and $t_0$, to simplify the notation.

Furthermore, we define a 2-dimensional time-dependent orthonormal basis, centred on $\boldsymbol{B}(t)$, and whose axes are the line connecting $\boldsymbol{x}_0$ to $\boldsymbol{B}(t)$, and its perpendicular. The components/coordinates of $\boldsymbol{g}_i(t)$ on this basis are hereafter denoted as $b_{i,\|}(t)$ and $b_{i,\perp}(t)$, as illustrated in the bottom panel of Fig. 2; they provide information on the spatial and temporal evolution of the

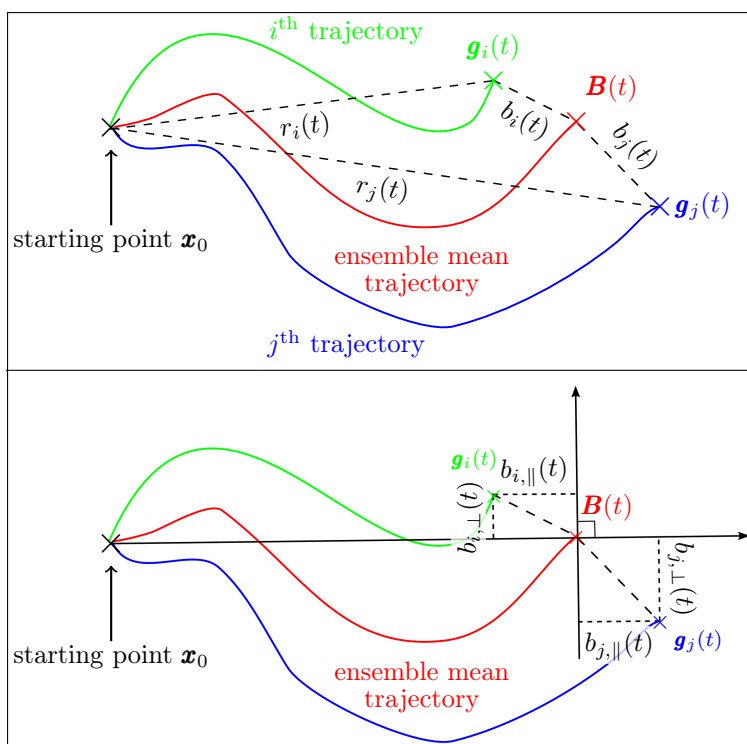

**Figure 2.** From a 12 member ensemble of simulated trajectories of a virtual buoy drifting during 10 days of which only two of them, denoted $i$ and $j$, are drawn, we represent the distances $r$, $b$ (top) and the coordinates $b_\parallel$, $b_\perp$ (bottom) for the virtual buoy $i$ and $j$ at time $t$.

ensemble spread and shape, and can also be used to look at how the virtual buoy positions are distributed around the ensemble mean over time.

With the individual $r_i$ and $b_i$ in hands, we compute basic, second-order, statistics. Let us consider their means, $\mu_r$ and $\mu_b$,

$$\mu_r(t) = \frac{1}{N} \sum_{i=1}^{N} r_i(t), \qquad \mu_b(t) = \frac{1}{N} \sum_{i=1}^{N} b_i(t), \tag{8}$$

and the standard deviations, $\sigma_{b_\parallel}$ and $\sigma_{b_\perp}$, of the components $b_\parallel$ and $b_\perp$,

$$\sigma_{b_\parallel}(t) = \sqrt{\frac{1}{N-1} \sum_{i=1}^{N} \left| b_{i,\parallel}(t) \right|^2} \quad \text{and} \quad \sigma_{b_\perp}(t) = \sqrt{\frac{1}{N-1} \sum_{i=1}^{N} \left| b_{i,\perp}(t) \right|^2}, \tag{9}$$

as our main quantities of interest in the analysis that follows. We note that the mean of $b_{i,\parallel}$ and $b_{i,\perp}$ are zero (being barycentric coordinates) and do not appear in the calculation of standard deviations. Throughout the rest of this paper, $\sigma_{b_\parallel}(t)$ and $\sigma_{b_\perp}(t)$ are only used to compute the ratio

$$R(t) = \sigma_{b_\parallel}(t) / \sigma_{b_\perp}(t), \tag{10}$$

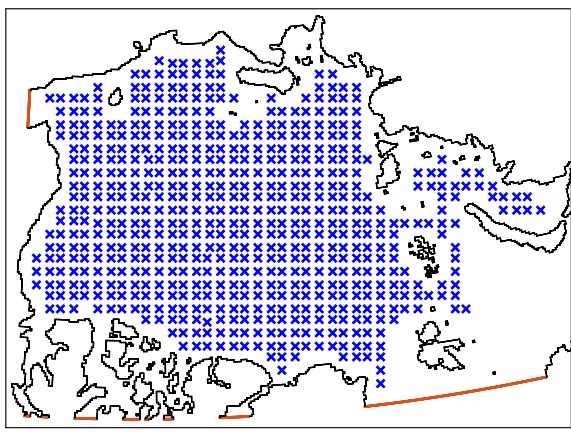

**Figure 3.** Maps showing the Arctic domain considered for this study. The red lines are open boundaries, while the black coastlines are closed boundaries. The starting points of the virtual trajectories simulated with the *neXtSIM* and FD models are represented by the blue crosses.

that provides a measure of the anisotropy of the ensemble spread of the virtual buoys positions around the barycentre $B$ of the ensemble.

It is finally worth observing that the two quantities, $r$ and $b$, provide complementary information: the former about the advective component of the motion, whereas the latter on its diffusive part. The ensemble mean distance from the starting point, $\mu_r$ is a statistical estimate of the distance travelled by an ice parcel according to the ice advection properties of the motion field, while $\mu_b$ is the (mean) spread relative to the aforementioned distance and accounts for the diffusion properties of the motion; see the top panel of Fig. 2.

### 3.2 Experimental setup

Our domain of study is the region covering the Arctic Ocean. While the coasts are considered as closed boundaries, open boundaries are set at the Fram and Bering Straits (see Fig 3).

The wind forcing is taken from the Arctic System Reanalysis (ASR) (Bromwich et al., 2012). This reanalysis product provides wind speeds and directions at 10 meters, every 3 hours, at a horizontal resolution of 30 $km$. No turning angle has been applied (see Table 1). For every 3-hourly wind field, we generate spatio-temporal correlated perturbations as described in Evensen (2003), and then add them to the ASR wind field. This procedure is identical to the one used to produce ensemble runs with the coupled ocean–sea ice model TOPAZ (Melsom et al., 2012) and constitutes the propagation step in the Ensemble Kalman Filter (Sakov et al., 2012). The method is designed such that the perturbed wind fields are keeping important physical properties, that is, the wind perturbations are geostrophic (gradients of random perturbations of the sea level pressure) and the

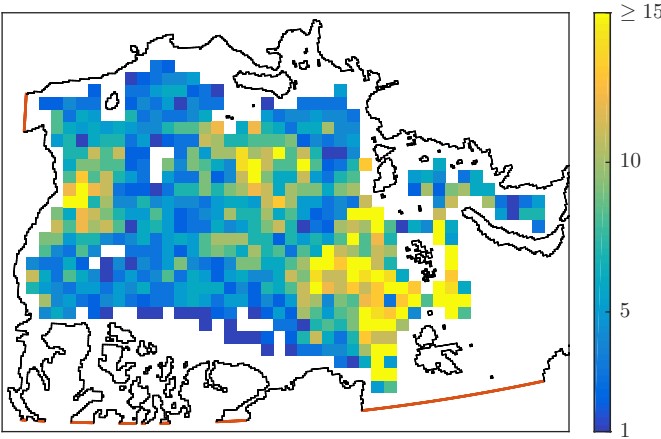

**Figure 4.** Spatial distribution of the number of occurrences of free drift events between 1 January 2008 and 30 April 2008. The temporal sampling frequency used is one day. These are the instances used for the optimisation of the air drag coefficient.

wind divergence is kept unchanged. They are built on random stationary Gaussian fields, with a Gaussian spatial covariance function, dimensionalised by the wind error variance and correlated in time. Time series of wind perturbations are assumed to be red noise. For our study, we used a decorrelation time-scale of 2 days, a horizontal decorrelation length scale of 250 $km$, and the wind speed variance as equal to 1 $m^2\ s^{-2}$. These values are identical to those used in Sakov et al. (2012) except for a

reduced wind speed variance to maintain a consistency with the ice rheology in *neXtSIM*. Indeed, a larger variance leads to an excess of ice breaking up beyond the physical behaviour expressed in *neXtSIM*.

Although the ensemble average of the perturbed u-, and v-components of the winds is equal by construction to the original winds provided by the ASR, the wind speed is positively biased. The value of the air drag coefficient ($C_a$ in Eq. (7)) had

previously been optimised in the *neXtSIM* model when forced by the ASR following the method presented in Rampal et al. (2016b), Sect. 3.2, and set to $7.6 \times 10^{-3}$. We applied the same method here to tune the value of $C_a$ so that the simulated ice drift compares best with the observed ice drift from the OSI-SAF dataset (Lavergne and Eastwood, 2015). The optimisation is carried out at all times between 1 January 2008 and 30 April 2008 but limited to the region where the ice is in free drift (see Fig. 4), that means, where the ensemble-average simulated ice velocity differs by less than 10% from the drift simulated by the

free drift model (see the FD model below).

Figure 5 shows the comparison, after optimisation of the air drag coefficient, between the observed and simulated ice velocities. As expected for a wind dataset positively biased in magnitude compared to the original one, we found an optimised value for the drag coefficient $C_a = 5.1 \times 10^{-3}$, lower than the one used in Rampal et al. (2016b) ($7.6 \times 10^{-3}$).

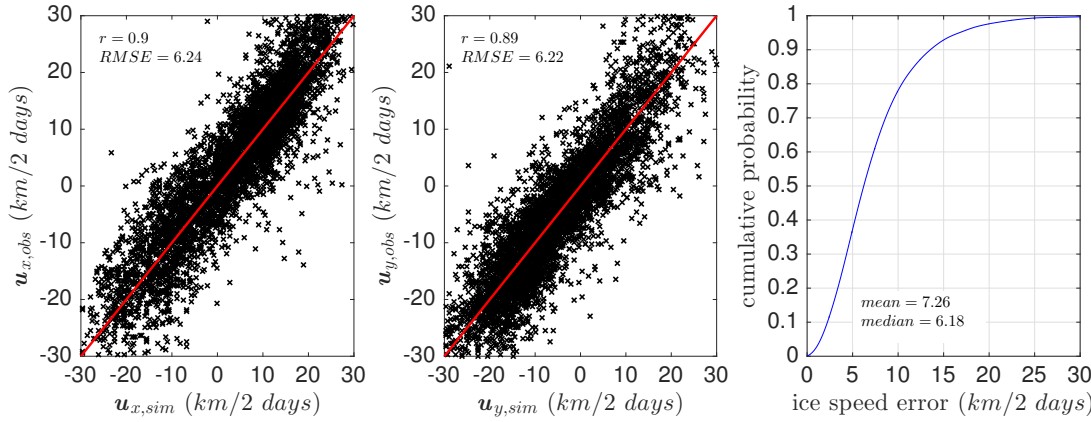

**Figure 5.** Scatter plots for the two components, $x$ (left) and $y$ (center), of the simulated (*neXtSIM*, x-axis) and observed drift (OSI-SAF dataset, y-axis) after the air drag optimisation procedure. The cumulative distribution of the ice velocity errors is shown in the rightmost panel.

The ocean forcing comes from the TOPAZ4 reanalysis (Sakov et al., 2012). TOPAZ4 is a coupled ocean-sea ice system combined with an ensemble Kalman filter data assimilation scheme assimilating both ocean and sea ice observations. In our simulations, we used the $30\ m$ depth currents, to which we apply a turning angle of 25 degrees, the surface temperature and salinity, and the sea surface height, all provided as daily means with an average horizontal resolution of $12.5\ km$, following
Rampal et al. (2016b).

Our analysis is based on two periods of the year 2008, respectively from 1 January to 10 May and from 1 July to 20 September, representative of the winter and summer conditions. We have intentionally studied them separately, because winter and summer are characterised by significantly different sea ice mechanical regimes, and therefore drift responses. During the win-
ter, the whole Arctic basin is covered by ice and its concentration is close or equal to 100%, that is, the internal stresses in the ice, and the corresponding $\nabla \cdot (\boldsymbol{\sigma} h)$ term in Eq. (2), becomes large and of the same order of magnitude as the wind drag term. As a consequence, the ice drift is (on average) much reduced. During the summer period, on the other hand, the ice concentration is lower and the ice pack does not generally reach the coasts, the ice internal stresses are much closer or equal to zero, and the ice drift closer to a *free-drift* state (see text below). We note however that the wind field perturbations are generated using
the same aforementioned procedure, for both the winter and the summer, and have thus the same spatial and temporal properties.

We ran a total of 13 simulations in the winter and 8 in the summer during successive, non-overlapping, 10 days-long periods. Limiting the length of the simulations to 10 days ensures that the sea ice state (thickness and concentration) remains as realistic as possible in the free-drift simulation, in which there are no physical limits to the amount of ridging and opening. The starting
positions are separated by $100\ km$ and cover the domain displayed in Fig. 3. All ensemble members start from the same initial

conditions extracted from a previous — deterministic — *neXtSIM* simulation by Rampal et al. (2016b) run without any pertur-bations of the winds. This concerns all sea ice variables : $h$, $h_{\mathrm{s}}$, $A$, $d$, $\boldsymbol{u}$, $\boldsymbol{\sigma}$. We ran an ensemble of 12 members, each of them forced by the perturbed wind generated as explained above. We performed (not shown) a convergence analysis of our results as a function of the ensemble size from $N = 3$ to $N = 20$, and observed a convergence from about $N = 10$ with only minor changes for $N \geq 12$, and are thus confident that $N = 12$ suffices to our purposes. From these ensemble runs we simulated a total of over 96 000 ($\simeq 8000 \times 12$) virtual buoy trajectories over the winter season, and over 38 000 ($\simeq 3200 \times 12$) trajectories over the summer season. This dataset was used to run the analyses described in Sect. 4 and presented at the 19[th] EGU General Assembly (Rabatel et al., 2017).

As already stated, we compared *neXtSIM* with the so-called *free-drift* model, hereafter referred to as FD, so that all simula-tions that follow have been carried out for the two models. *neXtSIM*, Eq. (2) with all terms in its right-hand-side included, is our reference model. The FD model is equivalent to *neXtSIM* except that it considers the following simplified version of the momentum equation in which the terms related to the sea ice rheology, the basal stress and the inertial term are neglected

$$0 = \boldsymbol{\tau}_{\mathrm{a}} + \boldsymbol{\tau}_{\mathrm{w}} - mf\boldsymbol{k} \times \boldsymbol{u} - m\boldsymbol{g}\nabla\eta. \tag{11}$$

In Eq. (11) the water and air drag forces, the Coriolis force and the gravity force due to the ocean surface tilt are balancing each other. The FD model is therefore analogous to the steady state drift of an object at the surface. We run the FD model with the same initial conditions as *neXtSIM* except that $d$ and $\boldsymbol{\sigma}$ are not used. The drag coefficient is also optimised for the FD model at a value of $3.2 \times 10^{-3}$, which is lower than for *neXtSIM*, as expected. The optimisation method used for FD is the same as for *neXtSIM* described above, except that the OSI-SAF drift vectors are used everywhere.

## 4 Results

In this section, the notations $<.>_W$ and $<.>_S$ correspond to winter and summer averages (i.e. over all the 13 and 8 simulation periods of 10 days) respectively. The notations $<.>_D$ correspond to the spatial mean over the domain. When considering both spatially and temporally averaged quantities, we use the notations $<.>_{W,D}$ or $<.>_{S,D}$.

### 4.1 Spatial patterns

Figures 6 and 8 show maps of mean drifting distance and spread (see the definitions of $\mu_b$ and $\mu_r$ in Sect. 3.1) of the virtual buoys after $t = 10$ days, averaged over the 13 (winter) and 8 (summer) successive simulations. Similar results are obtained for different time $t \in [0, 10]$ days (not shown). The pixels on the maps correspond to boxes of $100 \times 100 \ km^2$ centred on the initial positions $\boldsymbol{x}_0$ where the virtual buoys have been deployed at $t_0$.

Figures 7 and 9 are the counterparts of Figs. 6 and 8 and show the average wind speed (left panel) and ice thickness (right panel) for winter (Fig. 7) and summer (Fig. 9) respectively. Note that both figures are relative to *neXtSIM*, but the free-drift

wind speed is identical (same perturbations) and the ice thickness geographical pattern very similar; we have thus omitted to display them to avoid redundancy.

From Fig. 6 and 8 we see that *neXtSIM* gives a smoother response to perturbed forcing than the FD model in terms of mean advective drift $\mu_r$ and mean diffusive spread $\mu_b$, in both winter and summer. Indeed, we observe in *neXtSIM* a clear spatial coherency in both the advection and diffusion of the ice buoys over the domain that is almost absent in FD. We believe that this behaviour is related to the mean ice thickness pattern and, to a lesser extent, to the mean wind speed pattern (see Figs. 7 and 9 for winter and summer respectively).

For *neXtSIM*, the smallest values for the mean of $\mu_r$ and $\mu_b$ averaged over the winter time period are found in the area located north of Greenland and the Canadian Archipelago, which is where the ice is the oldest, thickest ($> 4\,m$) and mechanically the strongest, and where the winds are on average weaker as compared to the rest of the Arctic. On the other hand, in the surrounding Seas (i.e. Beaufort, Bering, Chukchi, Kara and Barents Seas from West to East), where the ice is thinner and the winds stronger, the mean of $\mu_r$ and $\mu_b$ are larger. Note that in summer these correlations or anti-correlations are even stronger, for example between the means of $\mu_b$ and the ice thickness (see Figs. 8 and 9).

For FD, the mean values of $\mu_r$ are correlated to the wind speed in winter and, to a lesser extent, in summer (left panels in Fig. 7 and 9). It is worth noting that, despite the presence of thick ice in the north of the Canadian Archipelago and low winds, the ice is still advected significantly, as opposed to what is obtained with *neXtSIM*. Moreover, the spatial pattern of the mean of $\mu_b$ shows no spatial coherence and resembles the random patterns from the wind perturbations. It is clearly visible in summer, while in winter the sea ice thickness field stays discernible. This may be due to the presence of the ice mass in Eq. 11.

In both winter and summer, the time-average response of $\mu_r$ and $\mu_b$ to wind perturbations is overall lower in *neXtSIM* than in FD (except in summer when $\mu_r$ is 7% larger in *neXtSIM*). This can be attributed to the ice rheology being turned on in *neXtSIM*, thus acting as an additional filter on the momentum transferred from the wind to the ice. In more details, it is interesting to note that the magnitude of the impact of the ice rheology is different whether we consider the drift distance by advection $r$ or the spread distance by diffusion $b$ and consider the winter or the summer. On average over the winter, $\langle \mu_r(t) \rangle_D$ and $\langle \mu_b(t) \rangle_D$ are respectively 21% and 52% lower in *neXtSIM* than in FD at $t = 10$ days, whereas over the summer, $\langle \mu_b(t) \rangle_D$ is 21% lower. This large difference between the two distances, especially in winter, is probably related to the high ice concentration making sea ice harder to break up, and keeps the members closer to each other. During summer, the ice is generally much less packed and the physical/dynamical differences between *neXtSIM* and FD have a lower impact. The 7% larger values of $\mu_r$ for *neXtSIM* are likely related to the optimisation of the air drag coefficient that returned a larger coefficient for *neXtSIM*.

As expected, for *neXtSIM*, we observe an increase of $\mu_r(t)$, of about $51\%$, and $\mu_b(t)$, of about $69\%$, in summer compared to winter. This behaviour differs drastically from the FD for which the values are nearly the same for both periods, and it is

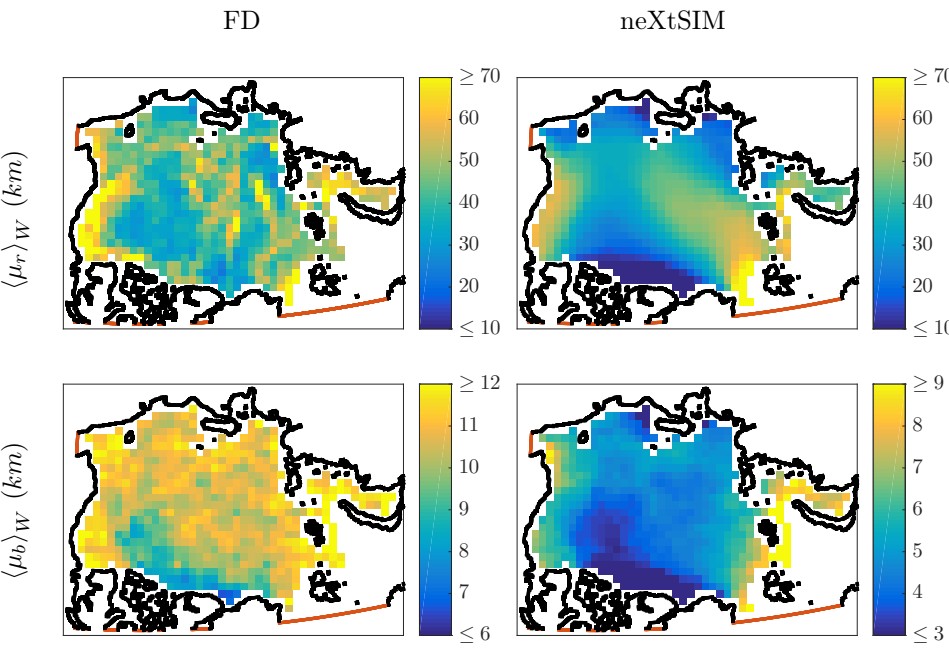

**Figure 6.** Mean over the winter period of $\mu_r(t)$ and $\mu_b(t)$ at $t = 10$ days. The calculated values are represented by coloured squares centred on the starting points $x_0$ shown in Fig. 3.

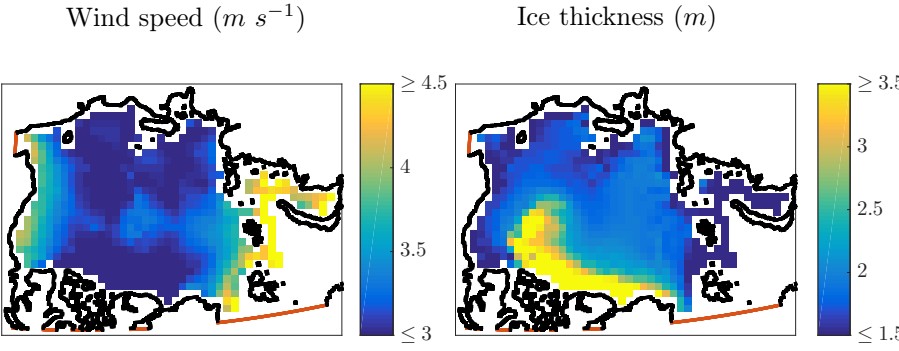

**Figure 7.** Winter average of wind speed and ice thickness. Both maps are from the *neXtSIM* simulations, but similar thickness field and exact same wind speed field are obtained for the FD simulations.

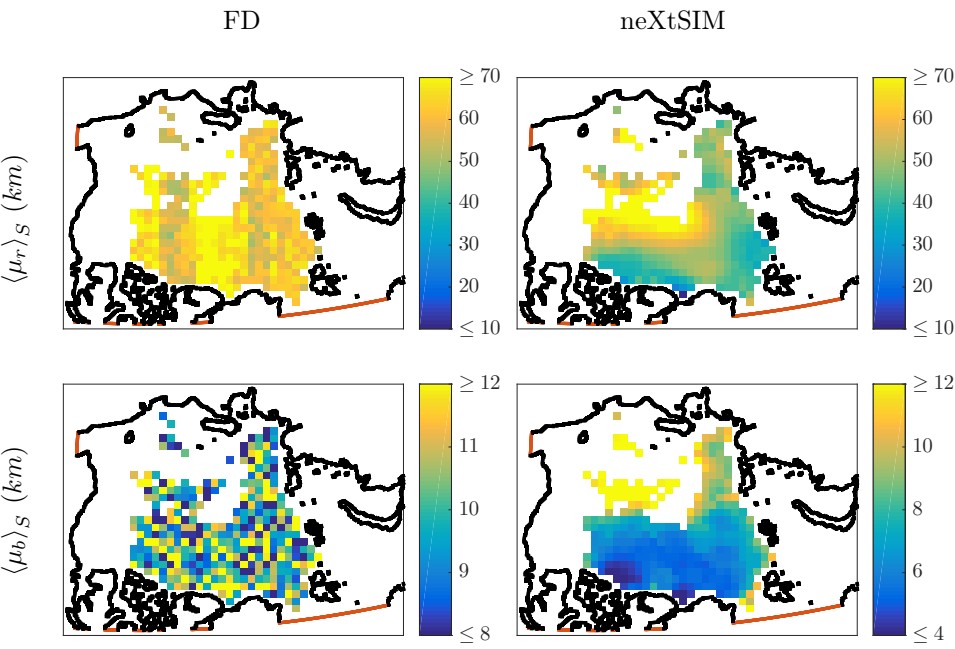

**Figure 8.** Mean over the summer period of $\mu_r(t)$ and $\mu_b(t)$ at $t = 10$ days. The calculated values are represented by coloured squares centred on the summer starting points $\boldsymbol{x}_0$ (not shown).

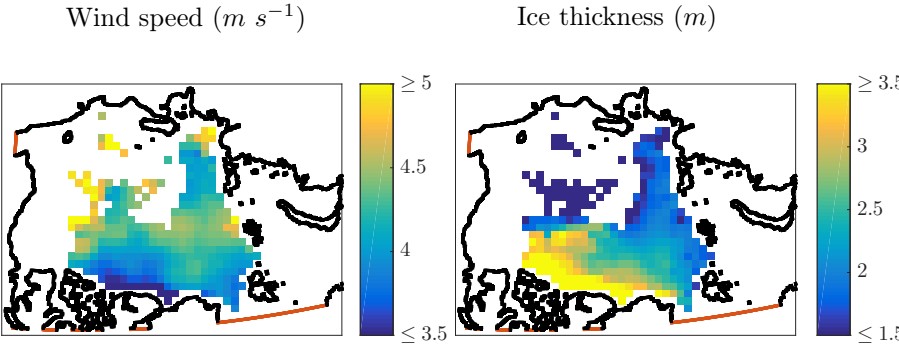

**Figure 9.** Summer average of wind speed and ice thickness. Both maps are from the *neXtSIM* simulations, but similar thickness field and exact same wind speed field are obtained for the FD simulations.

presumably related to the decrease in ice concentration due to the summer melting. The averaged sea ice concentration over the whole domain in winter is about $0.99$ while it drops to $0.83$ in the summer. In *neXtSIM*, this strongly influences the mechanical behaviour of the sea ice since the effective elastic stiffness $E$ depends non linearly on the ice concentration (see Eq. (5)). Assuming no change in the average level of damage of the ice, a drop by $15\%$ of the ice concentration between winter and

summer implies a reduction of $E$ by $96\%$. This reduction of $E$ leads in turn to a significant decrease of the internal stresses within the ice, thus lowering the term $\nabla \cdot (\boldsymbol{\sigma} h)$ in Eq. (2), which makes the buoys drift in *neXtSIM* closer to the ones obtained with the FD model.

     The absolute values of $\mu_r$ and $\mu_b$ obtained by our analysis reveal that the advection part of the motion is in general larger

than the diffusive part, independently of the season under consideration. In FD the ratio $\gamma = \mu_r(t)/\mu_b(t)$ at $t = 10$ days is about $4.5$. In *neXtSIM* though, the ice rheology is acting in increasing this ratio to $7$. However, this value presents a strong spatial variability depending on the local thickness and wind speed. Where both are large, $\gamma$ is large. For example, such areas are observed in the Fram Strait in winter ($\gamma > 10$), and in the central Arctic in summer ($\gamma > 12$). Where both ice thickness and wind speed are small, $\gamma$ is small. For example, this is the case around the new Siberian islands in winter ($\gamma < 4$), and close to

the ice pack edge in summer ($\gamma < 6$).

## 4.2    Spatial and temporal properties of the ensemble spread

Figure 10 shows the probability density function (PDF) of $b_\parallel(t)$ and $b_\perp(t)$ at $t = 10$ days for both *neXtSIM* and FD, and for winter and summer (see Sect. 3.1). The PDFs of $b_\parallel$ and $b_\perp$ for the FD case are almost identical, thus we chose to only display one curve (black dashed line). The first aspect to remark from Fig. 10 is that all distributions are uni-modal and symmetric,

suggesting that the 2D-shape of the ensemble is symmetric around its barycentre $\boldsymbol{B}$. However, we notice that the ensemble is anisotropic in *neXtSIM*, that is, the distributions of $b_\parallel$ and $b_\perp$ differ substantially, whereas it is close to isotropic in FD.

     Figure 11 shows the temporal evolution over 10 days of the Arctic averaged ratio $R$, Eq. (10), that defines the degree of anisotropy of the ensemble spread (1: isotropic; $> 1$: anisotropic). We observe on the one hand that $R$ is very close to 1 and

relatively constant over time in the FD model. On the other hand, it is systematically larger in *neXtSIM*, especially in winter, and it also displays a certain short-term variability. Here again, we encounter the peculiar effect of the *neXtSIM* mechanical response to the external forces, which is to break up and deform along fractures that are dispersing the different members of the ensemble along a preferential direction; such a behaviour cannot be reproduced by the FD. Note also that $R$ is as large as 2 within the first two days for *neXtSIM* in the winter, and then it decreases monotonically for $t > 2$, still remaining very

large (between $1.4$ and $1.6$ at $t = 10$ days). This reveals that the ice will first tend to move compactly along the initial fractures (identical for all members at $t = 0$) away from the origin, but it then starts to break and, after 2 days, the damage pattern becomes significantly different within each members leading to a more isotropic ice dispersion away from the barycentre.

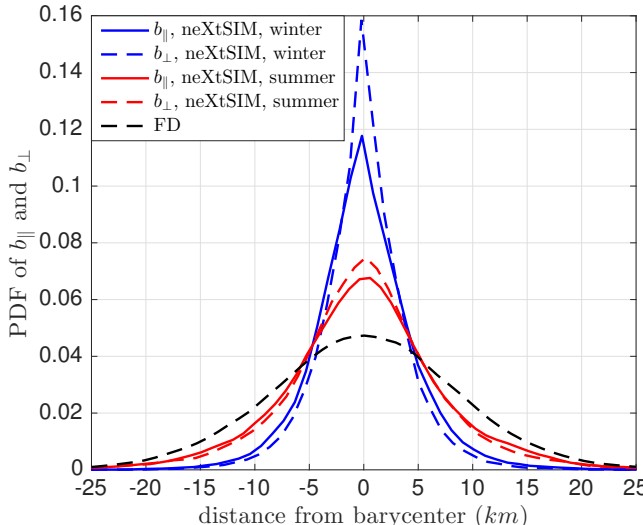

**Figure 10.** Probability density function of $b_\parallel(t)$ (solid lines) and $b_\perp(t)$ (dotted lines) at $t = 10$ days for *neXtSIM* in the winter (blue) and summer (red). The PDFs from FD are similar for summer and winter, and for $b_\parallel(t)$ and $b_\perp(t)$, and are therefore shown as a single black dashed line.

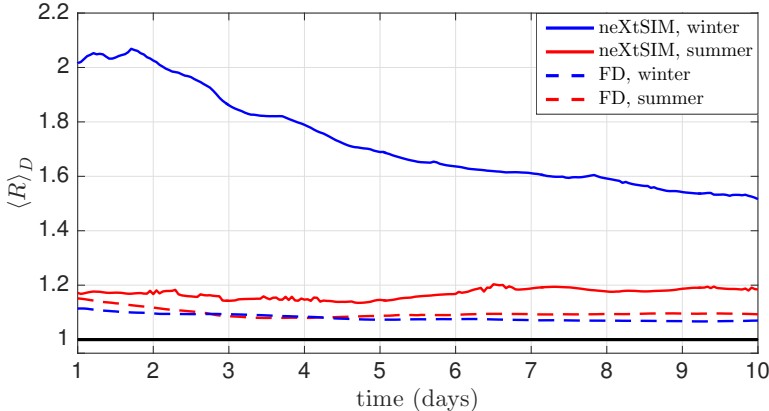

**Figure 11.** Evolution of the spatial mean of $R(t)$ from $t = 1$ to $t = 10$ days for the winter (blue) and summer (red) periods for *neXtSIM* (solid lines) and FD (dashed lines).

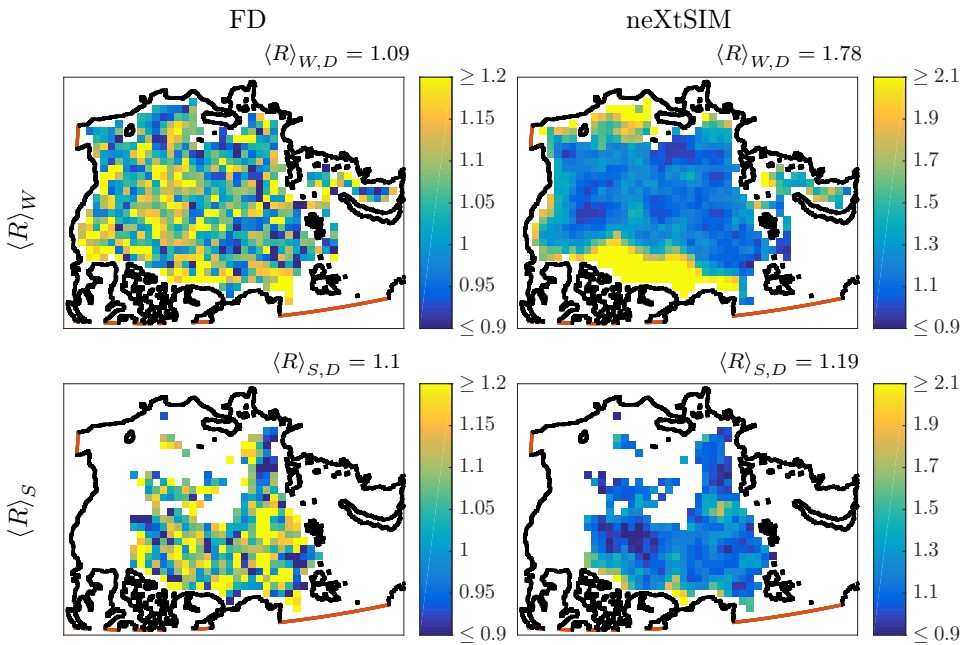

**Figure 12.** Means over the winter (top panels) and summer (bottom panels) of $R(t)$ at $t = 10$ days. The values are represented by coloured squares centred on the starting points $\boldsymbol{x}_0$. The spatio-temporal mean values (i.e. domain averaged and seasonally averaged) are calculated for each model and displayed above each panels for reference.

In Fig. 12, we show the maps of the $R(t)$ values computed for each ensemble of trajectories at $t = 10$ days. These values are represented as coloured squares centred on the starting point $\boldsymbol{x}_0$. We observe that highest degree of ensemble anisotropy ($R > 1$) is found north of Greenland and Canadian Archipelago, where the ice is the thickest and the ice drift and winds the lowest, in overall agreement with the interpretation of the temporal evolution of $R$ for *neXtSIM* in the winter, provided in rela-

5 tion with Fig. 11. Globally, we observe a high ($> 1.5$) anisotropy close to the coasts, that can be explained by the ice pressure that counteracts sea-ice motions towards the coasts (and the associated dispersion as well). In summer, large stretches of the coasts are ice-free and the increase of $R$ is less visible. This is in contrast to the pattern from FD. In absence of internal stresses, the pattern of the anisotropy exhibits no spatial coherence and is similar in both winter and summer periods. Furthermore, as already noticed from Fig. 11, the values obtained for *neXtSIM* are systematically larger (by about $65\%$) than for FD during

the winter whereas only $8\%$ larger during the summer. Yet, and remarkably, the values of $R$, and thus the anisotropy of the ice drift, for *neXtSIM* exhibit marked spatial correlations that are absent from FD.

Another important characterisation of the ensemble spread evolution can be set by looking at the variance of the distance $b$ between the virtual buoys and the barycentre $\boldsymbol{B}$ over time. The goal is to identify the diffusion characteristics of the ensemble,

which can be interpreted in the framework of the turbulent diffusion theory of Taylor (1921). Similar Lagrangian diffusion

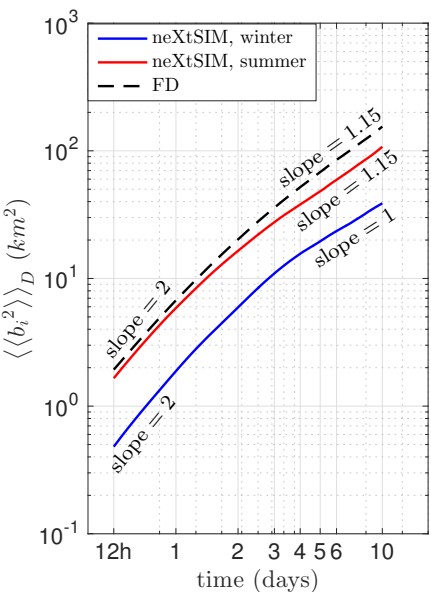

**Figure 13.** Spatial domain average of the variance of the distances $b_i$ as a function of time from $t = 12$ hours to $t = 10$ days, for the *neXtSIM* (solid) and FD (dashed) models, and for winter (blue) and summer (red). The results for winter and summer being identical in FD, only one curve is plotted (black dashed line). The Brownian regime (slope $= 1$) is reached by *neXtSIM* during the winter, while in the other cases, a *super-diffusive* regime is obtained (slope $= 1.15$).

analysis has been applied to study the regimes of diffusion of surface drifters in the ocean (e.g. Zhang et al., 2001; Poulain and Niiler, 1989), and more recently of buoys fixed to the ice cover (e.g. Rampal et al., 2009; Lukovich et al., 2015; Gabrielski et al., 2015; Rampal et al., 2016a). In the analysis performed here, the distance $b$ to the barycentre of the ensemble corresponds to the fluctuating part $m'$ of the motion $m$ in the so-called *Taylor's decomposition* $m = \overline{m} + m'$. Figure 13 shows the temporal
evolution of the ensemble average of the distances $b_i$ averaged over the Arctic domain $D$ calculated form the buoy's tracks simulated with *neXtSIM* and FD. We found that the ensemble spread follows two distinct diffusion regimes, one for small time $t \ll \Gamma$ and one for large time $t \gg \Gamma$ where $\Gamma$ is the so-called *integral time scale* (Taylor, 1921). In *neXtSIM*, the first regime we found for winter corresponds to the *ballistic* regime where $\left\langle \langle b_i{}^2 \rangle \right\rangle_D \sim t^2$, and the second to the *Brownian* regime where $\left\langle \langle b_i{}^2 \rangle \right\rangle_D \sim t$. These results are in agreement with the wintertime sea ice diffusion regimes revealed by applying the Lagrangian
diffusion analysis to the buoy trajectories dataset of the International Arctic Buoy Programme (Rigor, 2002) (Rampal et al., 2009) in wintertime, and shows that our experimental setup based on ensemble simulations forced by perturbed winds does not alter the capability of the *neXtSIM* model to reproduce these properties, as also shown recently in Rampal et al. (2016a) for the same winter. However, we note that the regime we obtain with *neXtSIM* for the summer 2008 is *super-diffusive*, with $\left\langle \langle b_i{}^2 \rangle \right\rangle_D \sim t^{1.15}$ for $t \gg \Gamma$, and therefore in apparent contradiction with Rampal et al. (2009) who found that sea ice follows
a same brownian regime in both winter and summer when averaging over the period 1979-2007. We suggest that this could

| Winter | Summer |
|:---:|:---:|
| 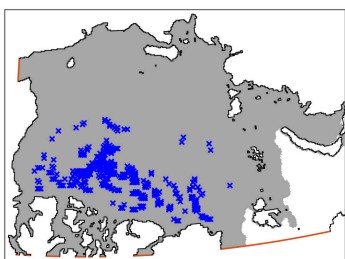 | 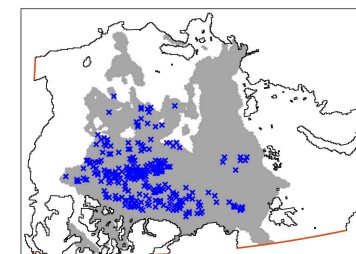 |

**Figure 14.** Maps showing the positions (blue crosses, 603 during winter and 344 during summer) of the IABP buoys trajectories dataset used in this study as starting point of the ensemble trajectory simulations performed with the *neXtSIM* and FD models. The grey area marks the presence of the sea ice during at least 10 consecutive days (the length of the simulations) during the winter and summer periods.

rather be the fingerprint of a change in the dynamical behaviour of sea ice in summer that occurred over the most recent years (including 2008), in which the rheology is playing a weaker role than it was in the 80's and 90's. This is also supported by the results we obtain here with the FD model that neglects the rheology, and which exhibits *super-diffusive* regime for 2008, regardless of the season considered.

### 4.3   Predictive skills of *neXtSIM* and of the FD models

We evaluate here how well the *neXtSIM* and FD models are able to forecast real trajectories in hindcast mode. As a benchmark, we compare the ensemble runs from both models to 604 (in winter) and 344 (in summer) observed trajectories from the IABP dataset. The simulated trajectories of both *neXtSIM* and FD are initiated on the same initial positions and at the same time as the IABP buoys, and are displayed in Fig. 14; the positions of IABP buoys are known every 12 hours. It is important to note that most of these buoys were deployed in regions of thick and compact ice, which drift is largely influenced by the sea ice rheology. Therefore, we expect the FD model to be less competitive than if the comparison data had been uniformly distributed across the Arctic.

As a metric for the models skill inter-comparison, we use the linear *forecast error* vector

$$\boldsymbol{e}(t) = \boldsymbol{B}(t) - \boldsymbol{O}(t), \tag{12}$$

defined as the distance between the observed IABP buoy position, $\boldsymbol{O}(t)$, and that of the ensemble mean, $\boldsymbol{B}(t)$ (see also Fig. 16). The components of $\boldsymbol{e}(t)$ onto the orthonormal basis centred on $\boldsymbol{O}$ (see Sect. 3.1, Fig. 2 and 16), read $e_{\parallel}(t)$ and $e_{\perp}(t)$. We complete this evaluation comparing results from both models with those from a single deterministic forecast in order to verify the advantage of probabilistic forecasts. In this case, we run *neXtSIM* with parameters found in Rampal et al. (2016b), except the air drag coefficient has been re-tuned to $C_a = 6.5 \times 10^{-3}$ and unperturbed winds. For this new air drag coefficient,

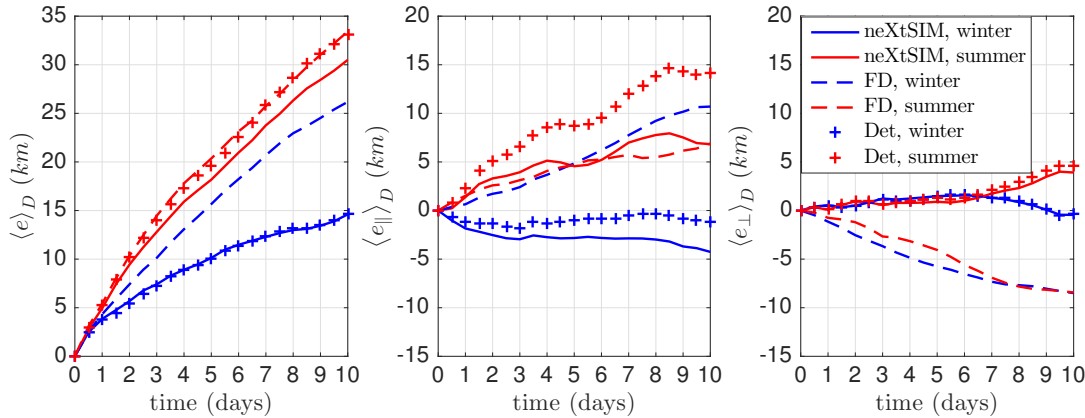

**Figure 15.** Mean of the absolute forecast error $\|e\|$ and vector components $e_\parallel$ and $e_\perp$ in the directions along and across the mean trajectory, as a function of drift duration. *neXtSIM* is represented by solid lines, FD is shown as dashed lines, and the deterministic runs with cross marks. Winter is in blue and summer in red. A positive $e_\perp$ represents a drift to the left of the trajectory.

the same optimisation process as the probabilistic case is used against the same observations.

Figure 15 shows the average norm of the forecast error, $\|e\|$, and of its components, $e_\parallel(t)$ and $e_\perp(t)$, as a function of time, for the experiments with *neXtSIM* and FD, and for both winter and summer. Results reveal that the forecast error is smaller
in *neXtSIM* than FD in both seasons. In winter, the error of the FD model grows almost twice as fast as the error of *neXtSIM*, up to 26 $km$ at day 10 compared to about 15 $km$ for *neXtSIM*. As already deduced from the results in the previous section, the mechanics underlying of the ice drift in *neXtSIM* and FD are similar in the summer, and this is reflected by the two errors being much closer to each other: the difference between the two increases slower, reaching $\simeq 3\ km$ after 10 days (see the left panel in Fig. 15).
The central panel in Fig. 15 shows a positive bias of the error in the along-drift component ($e_\parallel$) for both models and both periods, except for *neXtSIM* in winter which presents a negative bias. The general positive biases betray a too fast drift in the direction along the ensemble mean drift compared to the observations. Nevertheless, the bias for winter in *neXtSIM* is 2.5 times smaller than in the FD model, whereas both models perform similarly in the summer.

Finally, the right panel in Fig. 15 also reveals a bias of the error in the direction across the ensemble mean drift, yet substantially
weaker than in the previous case. For FD, $e_\perp$ still being negative for both periods corresponds to a drift too far to the right compared to the observations. This bias to the right could be further reduced by a separate tuning of the turning angle $\theta_w$ for the FD and *neXtSIM* models. Overall, we conclude that the performances are significantly better for *neXtSIM* in winter, but similar in summer and this would likely remain so even after optimal tuning of the turning angle.

Comparing to a single deterministic *neXtSIM* forecast, we note that the forecast error is close to the average of the probabilistic
run, although larger in summer reaching 34 $km$ at 10 days. The main difference with the probabilistic run is the poorer along-

drift component $e_\parallel$. Indeed, the error is closer to zero in winter and increases to 15 $km$ in summer.

In Hackett et al. (2006); Breivik and Allen (2008), Monte Carlo techniques are used to forecast the drift of an object on the ocean surface. They associate the density of trajectories at their end points to a density of probability and use them to define a
search area, within which the object is likely to be found. The search area is characterised by a surface centred on the ensemble mean and which size increases with the ensemble spread. The same methodology is followed here for forecasting the location of an object on drifting sea ice. In the context of rescue operations, the search area should be large enough to contain the actual position of the object, but not excessively large so as to keep the rescue operations time and resources affordable and efficient. The forecast system should therefore ideally yield a high probability to find the object in the search area, while keeping at the
same time the search area as small as possible for the cost-efficiency of the rescue procedure.

The probability to find a drifting object inside the search area, is referred to as the *probability of containment*, POC, and computed by counting the objects falling within the search area divided by the total number of objects. The POC may be interpreted as the ratio of the size of the search area to the square forecast error. Thus, a small forecast error compared to the
search area leads to a strong POC; conversely, a small search area (ensemble spread) compared to the forecast error leads to a poor POC.

In order to evaluate the probabilistic forecast capabilities of *neXtSIM* and the more classical FD model, the context of a *search and rescue operation* is adopted. We assume that an IABP buoy has been lost for 10 days: its initial position, $\boldsymbol{x}_0$ (see
Fig. 14), is assumed to be its last known position. The search area is then defined as the smallest ellipse centred on the ensemble mean position, $\boldsymbol{B}(t)$, encompassing all simulated members of the ensemble at time $t$. The main axes of the ellipse, $a_\parallel$ and $a_\perp$, are aligned respectively with the parallel and perpendicular directions from the initial position, as defined in Sect. 3.2, and illustrated in Fig. 16. Similarly to Eq. (10) an anisotropy ratio $R = a_\parallel/a_\perp$ can be defined: $R$ can be large due to the sea ice rheology. A search area defined in this way is increasing with the ensemble spread and contains 100% of the ensemble members.

Short of related literature for search and rescue in sea ice, we consider the values of open ocean search areas and POC found in Breivik and Allen (2008) and Melsom et al. (2012), as reference. These are respectively of the order of 1000 $km^2$ and 0.5, after 2 days of drift in the North Atlantic. We do not expect however a direct correspondence of these values to those of this section. First the sea ice is a solid, held together by the ice rheology, in particular in high concentration areas, so that
the ensemble spread is expected to be smaller than in the open ocean. Second, the currents in the North Atlantic are generally stronger than in the Arctic Ocean. Finally, the search areas may be more complex than just an ellipse; it may well be a set of disjoint areas, each one with an associated different POC (e.g. Abi-Zeid and Frost, 2005; Breivik and Allen, 2008; Guitouni and Masri, 2014; Maio et al., 2016).

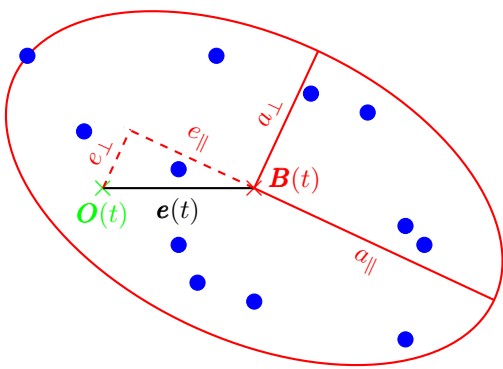

**Figure 16.** Illustration of the forecast error and the anisotropic search area. Blue dots represents the position of one member, while the barycentre of the ensemble (its mean) is $B(t)$. The observation $O(t)$ is in green and the forecast error is defined in Eq. (12). See text for definitions of the search ellipse and anisotropy ratio.

Figure 17 shows the evolution of the ellipse areas, averaged over all IABP buoys. The increase is nearly linear for both model configurations and seasons. After 2 days of drift in *neXtSIM*, the area does not exceed 100 $km^2$ in summer and not even half as much in winter. The area is larger in FD, and there is very little difference from winter to summer. The area for the FD is around 200 $km^2$ after 2 days and it reaches 500 $km^2$ after 5 days. The search area in FD is about 7 times larger than in
*neXtSIM* in the winter and 2.5 times larger in the summer. Therefore, even if the forecast errors are smaller in *neXtSIM* than in FD, its shrunk search areas lead to a smaller POC for *neXtSIM* than for the FD model (not shown).

On Fig. 18, we show the spread-error relationship for both periods and both models. The curves represent the spatial mean of the forecast error. Overall, the curves are above the black line, indicating that the forecast error is larger than the spread for
both models. The probabilistic forecast from both model during both period are therefore *too optimistic*: they underestimate the uncertainties of their forecast. However, it is interesting to note two properties of *neXtSIM*. First, for spreads larger than 4 $km$, the forecast error from *neXtSIM* becomes independent from the spread, unlike FD which errors grow monotonically. Second, for large spreads (greater than 3.5 $km$ in winter and 6 $km$ in summer) the curves from neXtSIM are consistently below those from FD and getting closer to the spread. Contrarily to the previous results, the FD and *neXtSIM* models behave very
differently in the summer.

The small values of the spread correspond to shorter forecast lead times (see Fig. 17) and these are the times when the *neXtSIM* model is still heavily influenced by its initial conditions of damage, as previously noted on the anisotropy ratio (Fig. 11). As the damage is irrelevant to the FD model, the initial error grows slower initially, but keeps growing while the
rheology maintains the errors closer to the spread in *neXtSIM*.

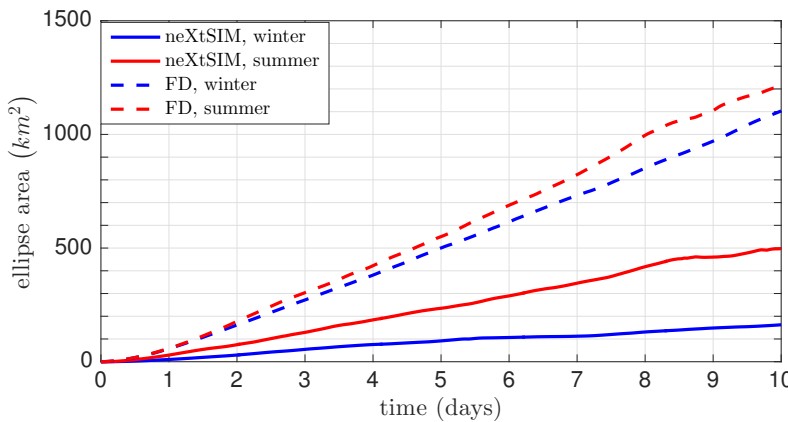

**Figure 17.** Time evolution of the averaged ellipse areas for *neXtSIM* (solid lines) and FD (dashed lines) in winter (blue) and in summer (red).

It should be no surprise that the two models underestimate the errors since this is a common behaviour of probabilistic forecast systems, but the differences of the shape of the spread-error relationships indicate that the two models underestimate the errors for different reasons: the *neXtSIM* ensemble is lacking spread during the initial times of the forecast but the asymptotic convergence of the spread to the errors tends to blame the constitution of the initial ensemble.

If one had considered the more linear spread-error relationship in the FD model alone, it would have been tempting to increase the variance of wind perturbation errors until a perfect match of the spread to the errors is obtained, but this would have over-tuned the variance of the wind and masked that the FD model suffers from unresolved physics.

### 4.4 Relevance for search and rescue operations

Whether a prediction model is too optimistic or too pessimistic may be equally problematic in view of search and rescue oper-
ations. In practice, the resources available for search and rescue operations are limited and only a given area can be covered, although the shape of the area (center and eccentricity in the case of an ellipse) may not influence the cost significantly. Thus, rather than looking at the size of the search area as estimated from the ensemble model prediction, the search-and-rescue operation can be posed as follows: for a given area to be searched, which model forecast gives the ellipse that is most likely to contain the object?

The ensemble forecast provides the expected position, $\boldsymbol{B}(t)$, and the anisotropy, $R(t)$ of the ellipse as defined previously, but the ellipse area is left free to grow homothetically from 1 $km^2$ to 3000 $km^2$. The POC increases then accordingly as the observed buoy position is more and more likely to fall within the ellipse. The dependency between the search area and the associated POC defines the so called *selectivity curve*, which makes possible a straightforward model comparison: the higher
the selectivity curve, the better the model's ability to locate the searched object. The selectivity curves also allow an immediate

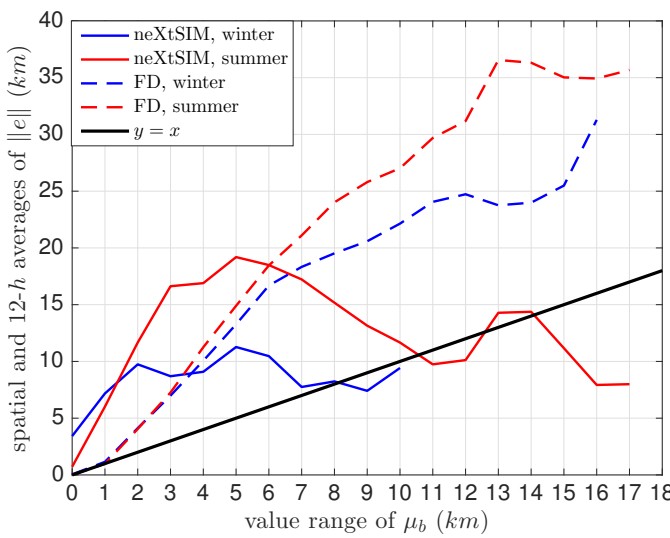

**Figure 18.** Spread-error relationship for 12-$h$ averages. The curves are based on the spatial mean of the forecast error shown as a function of the spread.

evaluation of the rate at which predictive skill is lost as a function of time.

For each time $t_0 + \Delta t$, with $\Delta t \in \{12, 24, 36, 48, \ldots, 10 \times 24\}$ hours, we compute the POC corresponding to search areas ranging from 1 $km^2$ to 3000 $km^2$ for both models and seasons. Results from *neXtSIM* (solid lines) and FD (dashed lines)

are shown at $t_0 + 1, 2, 3$ and 7 days in Fig. 19. In winter, for a given area, the POCs from *neXtSIM* are almost always above those from FD except in two cases: at $t_0 + 1$ day for search areas larger than 100 $km^2$ and at $t_0 + 2$ days for search areas larger than 500 $km^2$. If we neglect the anisotropy for these cases (i.e. consider circular search areas), the POCs from *neXtSIM* become larger than FD. This indicates that the strong anisotropy in *neXtSIM* is more a disadvantage for small time horizon and large search areas in this experiment. Otherwise for smaller areas, larger time horizons or in summer, considering circular or

ellipsoidal search areas makes no difference (not shown). As long as the drift is longer than 3 days, the selectivity curves of *neXtSIM* are systematically above FD. Whereas, in summer, for any time horizon and any POC, the results are very similar with a faint advantage to *neXtSIM*. When comparing to POCs of ellipses centered on forecasts from a deterministic *neXtSIM* run (not shown), the results are identical in winter and poorer with the deterministic run in the summer.

For both periods and both models, all curves exhibit a sigmoid shape with an inflexion point, which position depends on the time horizon (higher POC and larger search areas for longer drift duration). For a 7 days drift in winter and a POC equal to 0.5, the area is around 300 $km^2$ with *neXtSIM*, while it reaches 1000 $km^2$ in FD. In the summer, a larger area is necessary to obtain the same POC for both models. For a given search area, the gap between the POCs from *neXtSIM* and FD seems independent

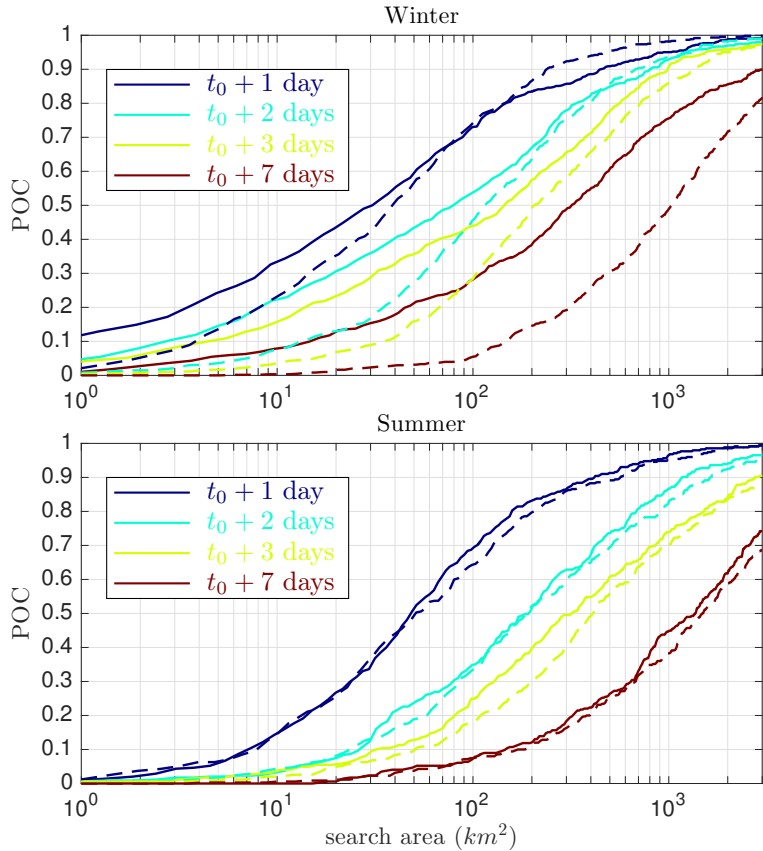

**Figure 19.** Time evolution of POC according to the search area for *neXtSIM* (solid lines) and FD (dashed lines) in winter (top) and in summer (bottom) for different time horizons.

from the drift duration in summer, whereas in winter it increases with the time prediction horizon. It is interesting to note the lowermost value of the POC for small areas in winter, which remains above 0.1 for *neXtSIM*. This could be a consequence of the capability of *neXtSIM* to simulate immobile ice, while the FD ice is always in motion with the winds and currents.

How do the different models perform for different forecast time (i.e. drift duration)? To answer this question, we study the
5  time evolution of the difference between the *neXtSIM* and FD POCs: when this difference is positive/negative *neXtSIM*/FD is outperforming FD/*neXtSIM*. The POC for both models is evaluated for a fixed search area - a vertical section across the selectivity curves - equal to 50 $km^2$ in winter and 175 $km^2$ in summer, and the results are shown in Fig. (20). The chosen values of the search areas, 50 and 175 $km^2$, correspond to the mean ellipse areas based on the ensemble spread from *neXtSIM* after 3 days, averaged over the IABP dataset (see Fig. 17) respectively in winter and in summer. Figure 20 reveals that, after
10  2 days in the winter, the POC of *neXtSIM* is larger by about 0.2 than the POC of FD; most remarkably, such a substantially improved skill is then maintained almost stably up to the last day of simulation (10 days). During the summer, the POC of

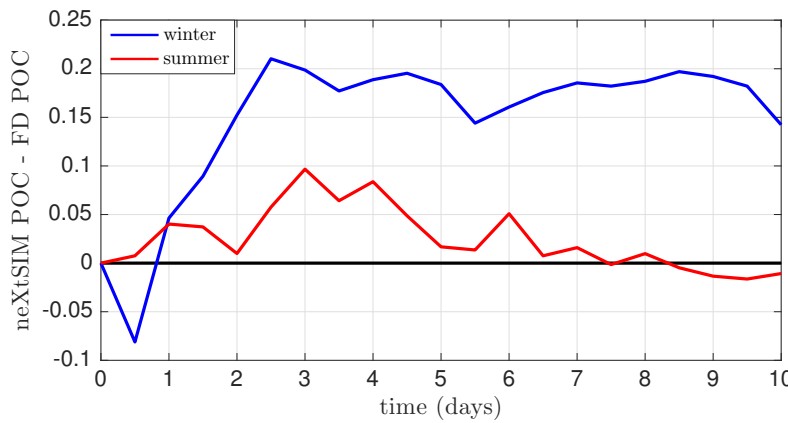

**Figure 20.** Time evolution of the POC difference between *neXtSIM* and FD for a search area equal to 50 $km^2$ in winter (blue) and equal to 175 $km^2$ in summer (red).

*neXtSIM* is also generally higher than the one of FD, but the difference is half of the one observed in the winter. Furthermore, after the 3$^{rd}$ day, the difference between the two models decreases to vanish completely between day 8 and 9. The fact that most of the superiority of *neXtSIM* is found during winter is logical and should be no surprise given that during the summer the ice mechanics in the two models are similar.

The negative values for lead time shorter than 1 day in winter is again likely caused by the initialization of the *neXtSIM* ensemble and another reason to constrain its initial anisotropy to observations.

## 5   Discussions and Conclusions

The ensemble model sensitivity experiment carried out with *neXtSIM* and with an FD model reveals the prominent role of the rheology, which marks the key difference between the two models. On average over the whole Arctic *neXtSIM* is less sensitive to the wind perturbations than the FD, albeit large seasonal and regional differences are observed. This is exemplified by the imprint of the ice thickness field in the ensemble spread from *neXtSIM* and the much smaller sensitivity of *neXtSIM* in winter than summer, in contrast to the FD model (Fig. 6 and 8). Both aspects point clearly to the role of the rheology which accounts for the ice thickness and compactness. This behaviour should be expected to hold also for other sea ice rheologies than the elasto-brittle.

The two models have been tuned on different observations of ice, seen as in free drift by each model, so that the different performances originate by the differences in the resolved model physics at their best performance. The diffusion regimes of *neXtSIM* and FD are very different in winter (Fig. 13): the offset between the curves indicating differences of sensitivity, and

the slopes indicating different rates of increase and thus sea ice diffusivity. The expected differences between summer and winter are only represented when the rheology is turned on.

Due to the dispersive properties of the sea ice, the shape of the ensemble of simulated buoys positions is generally anisotropic. Such anisotropy is a signature of the underlying mechanism that drives the dispersion of the members, which is the shear deformation of the ice cover along active faults/fractures in the ice. This mechanism is missing in the absence of rheology (like in the FD model) and represents a clear strength in principle for the elasto-brittle rheology in *neXtSIM*, although with the present ensemble initialization, it did not prove to be a practical advantage. Other rheological models, such as the Elasto-Viscous Plastic model, also present some degree of anisotropy (Bertino et al., 2015), although the two models have not been compared in the same conditions.

The performance of the two models differs significantly when forecasting the trajectories of IABP buoys. The ensemble mean position errors are larger in the summer (5 $km$ after 1 day and 12.5 $km$ after 3 days drift for *neXtSIM*, about 16% below the FD results), and consistent with the values reported by Schweiger and Zhang (2015) (RMS errors of 6.3 $km$ and 14 $km$ respectively, but using different time periods). The corresponding errors are smaller in winter, especially for *neXtSIM* (25% smaller than FD) and down to 4 $km$ for a 1-day drift and 7.5 $km$ for a 3-days drift. These values seem competitive compared to the year-round average RMS error of 5.1 $km$ per day in the TOPAZ4 reanalysis (Xie et al., 2017), even though the ice drift measurements are assimilated in TOPAZ4 (Sakov et al., 2012). The RMS errors of the free drift model in Grumbine (2003) also seem to be higher than 5 $km$ per day.

The model sensitivity to wind perturbations has been evaluated, yielding (for 10 days drift) a spread from 5 to 10 $km$, for winter and summer respectively, but this is smaller than the corresponding errors (15 $km$ from the barycentre to the observations in Fig. 15). Still, since the diffusion regime is respected (at least in the winter), we are confident that the spread simulated by the model is physically consistent. Other methods for perturbing the winds should be tested to remove the super-diffusive behaviour in summer however.

To futher improve the spread-error relationship, alternative sources of errors should be considered such as, for example, model initial conditions and forcings (ice thickness, concentrations, damage, ocean currents). Since the errors are increasing faster in the first days of the simulations, the more likely source of local and short-term errors lies in the position and orientation of the sea ice fracture network, which is left unconstrained in any of the experiments presented here.

Although we would expect an increase of the ensemble spread if the ice thickness, concentrations and ocean currents had been taken into account in the ensemble initialization, we do not believe it would lead to a much larger spread, especially in the winter. We suggest instead that, in the perspective of efficient sea ice forecasting, major efforts should be directed toward assimilating the observed fractures (as of satellite images). The assimilation of fracture (as objects rather than quantitative observations) represents a priori a challenging avenue in terms of data assimilation, which traditionally deals with quantitative

scalar or vector observations, however we envision that the damage variable in *neXtSIM*, showing localized features, can be constrained quantitatively to deformation rates as derived from observed high-resolution ice motions and serve as "object assimilation".

In spite of the biases, the selectivity curves indicate that a probabilistic forecast using *neXtSIM* is largely more skilful than the traditional free drift model, and it has the larger potential for practical use in search and rescue operations on sea ice. Since the Arctic is not easily accessible, forecast horizons of 5 to 10 days are probably the most relevant for logistical reasons. On those time scale, the differences of POC shown in Fig. 20 indicate that the free drift model gives a poorer information in winter because of the biases in the central forecast location and the lack of anisotropy, while in the summer the use of an elasto-brittle

rheology is only marginally advantageous. The comparison of deterministic versus probabilistic forecast gives, as expected, an advantage to the average of the probabilistic forecast, although it is rather small and surprisingly more important in the summer, although the model non-linearities are stronger in the winter.

    The physical consistency of the ensemble sensitivities is a necessary condition to the success of ensemble-based data assimi-

lation methods (Evensen, 2009), which constitutes one of the follow-up research direction the authors are currently considering. Combining the modelling and physical novelty of *neXtSIM* with modern observations of the Arctic is seen as a major asset for forecast and reanalysis applications.

    Besides the potential use of observations of fractures, as mentioned above, which is indeed another unique advantage of

models such as *neXtSIM*, ice drift data are also crucial. Observations of ice drift are still seldom used for data assimilation, and when it is the case, the success is limited by the lack of sensitivity of the sea ice model (see, e.g. Sakov et al., 2012). Nevertheless, the main fundamental issue related to the use of data assimilation, and particularly ensemble-based methods, stands in the nature of the Lagrangian mesh of *neXtSIM*, which also include the possibility of re-meshing (Rampal et al., 2016b). This feature, while essential to the skill of the model in describing the mechanics of the sea ice with great details,

represents a challenge in developing compatible data assimilation schemes, as the dimension of the state space can change over time when these re-meshing occur. This problem has recently attracted attention in the data assimilation research community (see, e.g. Bonan et al., 2016; Guider et al., 2017; Carrassi et al., 2017) and it is also a main area of on-going investigation of the authors, following the present study.

*Author contributions.*   The sensitivity analysis has been implemented and performed by MR. The results have been analysed by MR, PR, AC

and LB. The manuscript has been written by MR and PR, then reviewed and improved with inputs from all authors.

*Competing interests.*   The authors declare that they have no conflict of interest.

*Acknowledgements.* M. Rabatel, P. Rampal, A. Carrassi and L. Bertino have been funded by the Office of Naval Research project *DASIM* (award N00014-16-1-2328). P. Rampal, A. Carrassi and L. Bertino also acknowledge funding by the project *REDDA* of the Norwegian Research Council. C.K.R.T. Jones was supported by the Office of Naval Research (USA) under grants N00014-15-1-2112, and N00014-16-1-2325. The authors are grateful to J. F. Lemieux, H. Goessling and one anonymous reviewer for their insightful comments that helped improve the manuscript.

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
