# Peer review of "Impact of rheology on probabilistic forecasts of sea ice trajectories: application for search and rescue operations in the Arctic"

_The Cryosphere, 2017_

## Referee Comment (RC1) · J.-F. Lemieux (Referee) · 14 Nov 2017

**Review of 'Probabilistic forecast using a Lagrangian sea ice model: application for search and rescue operations' by Rabatel et al.**

In this paper, the authors implement a probabilistic forecast capability for the neXtSIM sea ice model and compare these forecasts to the ones of a free-drift (FD) model. The ensemble of sea ice forecasts is produced by forcing the sea ice model with perturbed wind fields. They also investigate the use of these probabilistic forecasts for search and rescue applications.

It was a pleasure to read this paper. I found many ideas inspiring. I am not very familiar with search and rescue applications and found this section very interesting.

I am not really surprised to see that neXtSIM does better than the FD model (especially in winter). In fact, I would like to suggest a different way to present these results. Instead of saying you compare two models, you could say you study the impact of rheology on probabilistic forecasts. The title of your paper could be: Probabilistic forecast using a Lagrangian sea ice model: impact of rheology. This is just a suggestion.

The paper is in general well written. There are a lot of interesting results. There are, however, a few things that need to be clarified. I also suggest a few additional things that could be worth investigating. Please see my comments below:

**1. Major comments**

1) You need to give more details on how you initialize the forecasts. Do you use fields (h, hs, A, d, u) from the previous forecast? And how do you deal with the FD model? I guess you use the thickness field from neXtSIM as the thickness field from a model without rheology would be completely unrealistic. Please relate that to the caption in Fig. 6.

2) I understand why you neglect the rheology term for your FD model. However,

what is the justification for neglecting the inertial term?

3) Fig 2. and p. 9 line 3: How do you define FD 'events'? Concentration threshold?

You have optimized $C_a$ for neXtSIM. Your conclusion (p. 23) says you have done the same thing for the FD model. This should be mentioned and clarified earlier. I also suggest you give the $C_a$ value you obtained for the FD model.

4) You often discuss spatial correlations between certain fields (e.g. Fig 7 and 8). You relate these high correlations to the rheology and the thickness field. I think it would be a good idea to show maps (winter and summer) of the effective elastic stiffness as it is more representative of the 'strength' of the ice cover than just the thickness field. I am also wondering what is the effect of the pressure term? My impression is that the effective elastic stiffness gets very small in summer because the ice is so damaged and cannot heal so that the pressure term plays an important role.

5) In the comparison of the predictive skill of the model with and without rheology, you look at the error of the barycenter. I think you could also discuss whether the error e(t) of the barycenter is smaller than the one of a single deterministic forecast (no perturbation to the wind). Even if it is not the case, the probabilistic forecasts with its spread would still give important information...

I would also be curious about the following experiment...what happens if you move your virtual buoys with the persisted initial velocity of the observed buoys (see Hebert et al. 2015). At what lead time is the ensemble of neXtSIM better than the persisted observed initial velocity? I guess this could give you some indications about the quality of your forcing field.

**2. Minor comments**

1) Overall the english and the text is very good. There are a few typos. Here is a list of some of them: p.1 line 14, p.4 line 4, p. 11 line 29, p. 20 line 5, p. 22 line 29, p. 25 line 28.

2) p.2 line 5: Add 'sea ice' before 'forecasting systems'.

3) p.2 line 5: Note that RIPS is no longer in operations and has been replaced by the coupled Regional Ice Ocean Prediction System (RIOPS). It would be better to rephrase. The references for this new system are Lemieux et al., 2016 (the paper you already cite) and Dupont et al. 2015:

   A high-resolution ocean and sea-ice modelling system for the Arctic and North Atlantic oceans.

4) p.3 line 3: remove 'advanced'...Just say what it is.

5) p.3 line 9: Coon et al., 1974 modeled sea ice as an elasto-plastic material...please rephrase.

6) p.3 line 18: 'sea ice responds in a linear way' is vague. Please clarify what you mean by that.

7) p.3 line 20: You could add '(due to the limited number of observations)' at the end of this sentence.

8) p.3 line 27: Change 'full complexity of the present version' by 'the latest model developments' .

9) p.4 line 9: Change 'spatial' by 'spatially'.

10) p.4 line 20: Change 'refreezing' by 'freezing'.

11) p.4 line 23: What do you mean by 'effective'? Grid cell mean values?.

12) p.4 eq. 2: I am not familiar with this formulation of the vector product for the Coriolis term...Don't you want to use the common formulation with the 'x'?

13) p.6 line 8: Add 'virtual' before 'buoy'.

14) p.6 line 20: I think you need to divide by N in the equation for B(t).

15) The second figure you refer to is Fig.4 (p. 8 line 4). Please change the order.

16) p. 8 line 5: Are these 10 m winds? Please specify this and mention the turning angle you use (maybe also for the ocean currents).

17) p. 9 line 3: Remove 'state-of-art'...Just say what it is.

18) p. 9 line 17: 'Dominant' is a bit confusing here because it sounds like it is the largest term in the momentum equation (the wind stress is usually the largest one). Please rephrase.

19) p. 10 line 3-5: Why are thermodynamics an issue? You have a thermo model, right?

20) p. 10 last line: Add 'steady state' before 'drift'.

21) Fig.5: It is difficult to see the coherency between the neXtSIM panels because the lower panel is almost only blue. Can you improve the colorscale so that we can see better the difference? (same idea for Fig. 7)

22) p. 11 line 25-30: You mention correlations between spatial fields. Is it just by looking at the figures or you actually calculated spatial correlations?

23) p. 14 line 1: Clarify what you mean by 'the response'...ice velocity?

24) p. 15 line 3 and elsewhere: Is 'on another hand' a correct expression? Is it better to use 'on the other hand'?

25) Fig. 13: How do you define the mean sea ice coverage (A=15% contour)? Looking at these two panels, as all the buoys are in regions of thick compact ice, it is kind of obvious that neXtSIM will do better than FD in this experiment. In other words, the FD model would do better if the buoys were uniformly distributed. I would add a sentence to mention that.

26) As you calculated the POC for Fig. 17, I suggest you give the exact definition of the POC in eq. 13 instead of saying that it is proportional to...

27) Fig. 17: same idea as before, what happens if you use the deterministic forecast instead of the barycenter? Do you get a real benefit from the ensemble forecast for the time evolution of the POC?

28) Please rephrase the last sentence of p. 22.

29) p. 23 line 3: replace 'sensitivity' by 'sensitive'.

30) p. 25 lines 7-9: I understand what you mean but I find that the two sentences ('Still it is the wind...' and 'we suggest instead...' ) kind of contradict each other. If the wind is the key player, efforts should be made to improve the forcing winds (by improving the assimilation and forecasts of the atmospheric model). Just rephrase a bit. By the way I like the discussion about assimilating sea ice fractures...Interesting.

Congratulations for your paper.

Jean-François Lemieux

---

## Referee Comment (RC2) · H. F. Goessling (Referee) · 17 Nov 2017

**Summary**

Rabatel et al. investigate the drift behaviour of virtual buoys in a sea-ice model with elasto-brittle rheology and a Lagrangian unstructured mesh. They construct ensembles by driving several realisations with different spatio-temporally correlated wind perturbations added to reanalysis-based atmospheric forcing. They compare results to a simplified free-drift version of the model to assess the impact of the sea-ice rheology. The authors compare simulated ensembles of trajectories with observed drift from IABP data. They show that the model with rheology better captures anisotropic drift

dispersion and, despite being underdispersive, can help to define search and rescue areas with increased probabilities of containment compared to the free-drift model, in particular in winter.

The paper is well written and the quality of the research and presentation is high. I have a number of remarks that might help clarifying some aspects, including quite a number of technical corrections and comments. Overall, I clearly recommend the manuscript should be accepted, after minor revisions.

(Note that I have not read the other review prior to writing this review to ensure an independent assessment.)

**Specific comments**

Concerning terminology, I think it would be worthwile to clarify that this study is using the term "forecast" not in the sense of forecasting actual future trajectories, where the future evolution of the system is becoming more and more uncertain over the forecast lead time through chaotic error growth, but in a slightly different way where the future evolution of the most chaotic component - the atmospheric forcing - is approximately known. Of course uncertainties are introduced in another way, namely through perturbations of the atmospheric forcing, but still the underlying synoptic evolution is the same in all ensemble members. I'm not trying to say that this is not worthwile doing; in particular, one can imagine search & rescue applications where one aims to find the current position of a target that got lost 10 days ago, so one could run a "forecast" system like the one used here to "nowcast" the current position using near-real-time atmospheric (re-)analyses. And, obviously, one could also use actual atmospheric forecasts to drive the model, but that is not done in this study, so I recommend to just clarify this.

P2L33: "departing from independent in situ drifting buoys, and compare them with real observations"; Does "in situ drifting buoys" not refer to the "real observations"? Please clarify.

P3L28-29: "the impact of some mechanical parameters on the ice deformation can still be considered as valid"; the "some" sounds very vague, could you be more specific?

Eq1: If I am not mistaken, this holds only when h and h_s are the "effective" (grid-cell averaged) thicknesses, correct? And for A it stops holding for A close to 1, in particular if there is a lot of damage where there can still be considerable convergence despite A=1 (even with the "pressure term"), right? This would deserve some clarification.

Eq5: It may help to mention what value is used for alpha (probably -20 as in Rampal et al. 2016?) so the strongly non-lineas dependence on A becomes obvious.

P5L24-28: To me this paragraph sounds very vague; could the authors be more specific on what inputs and outputs are considered?

Eq8+9: It might be worth noting that the means of "b_i,II" and "b_i,L" are zero and thus omitted in Eq9. It might also be worth pointing out that mu_b contains basically the same information as "b_i,II" and "b_i,L" (except the directional information); they do not relate to each other like the first and the second momentum of a distribution (which is not stated, but at least I was confused at first).

Fig2: What data and analysis is this figure based on? And what temporal sampling frequency is used to detect "events", e.g., one day?

P9L17: "the internal stresses in the ice, and the corresponding Grad(sigma h) term in Eq. (2), becomes very large and dominant"; Would it be more precise to say that it almost completely balances the other forces (so that the acceleration (and speed) becomes very small)?

P10L6: "We ran an ensemble of 12 members, each of them forced by the perturbed wind dataset generated as explained above"; If I have not overseen some important detail, there is some information on the experimental setup missing. In particular, how are the sea ice and ocean in the different members initialised? Is there one single "reference run" from which the ensembles are brached off, with all members keeping the

same initial sea-ice/ocean state? If so, does the reference run also have perturbations to the winds (and accordingly uses the re-tuned parameters)? Or are there just 12 simulations overall, covering the whole time period, so that the "initial" sea-ice/ocean states are different between the ensemble members? The latter doesn't seem to be the case as you speak of inidividual simulations in P10L3. Also, P25L5-6 seems to hint that indeed the initial states are identical. In any case I have the impression that the question of whether or not the initial sea-ice states are identical is very important for the interpretation of some of the results (see below), so I think this should be described very clearly.

P10L9: "8000 virtual buoy trajectories over the winter season"; Is this the number of ENSEMBLES of buoy trajectories? For individual trajectories I would expect a larger number, given the approximate number of initial positions in Fig4 and the number of 10-days periods.

Eq11: While you can certainly say that the omitted presseure term, as the stress term, belong to the rheology, the omitted tau_b could also be mentioned.

P10L19: "The FD model therefore mimics the drift of a buoy at the surface of the ocean."; I would think that this is not really the case because the drag coefficients would be quite different (in particular on the water side due to turbulent momentum transfer between deeper layers and the surface water surrounding the buoy)?

Fig5+7: i) I do not understand why the sea-ice thickness pattern is so clearly visible in the dispersion strength (mu_b) for the free-drift model where the rheology shouldn't play any role; could the authors comment? ii) I suggest to use the same colour scales for the two bottom panels so that the difference in mu_b becomes even more obvious.

P14L1: "In both winter and summer, the response to wind perturbations is overall lower by 35% in neXtSIM than in FD"; Where does this number come from? I would have thought that the difference of mu_b in neXtSim versus FD would quantify "the response to wind perturbations", but those are reduced by 63% and 39% in winter and summer,

respectively (as statet in P14L5-6), so that doesn't fit. Could you please clarify? (Also at the beginning of Sect.5)

P14L21-27: Is the assumption correct that the values found for the ratio mu_r/mu_b should scale with the strength of the wind perturbations? If so, this might be worth mentioning.

P15L8-9: "This reveals that the ice will first tend to move compactly along the wind direction away from the origin, but it then starts to break and depart from the barycentre"; First, the wind directions felt by the different ensemble members differ instantly after the initialisation, right? So, moving compactly along the wind direction would imply a slightly different direction for each member from the very beginning. Second, the ice is "broken" (i.e., has fractures) already at initial time, right? Third, and maybe more importantly, I think that the interpretation of the decreasing anisotropy might depend strongly on the initial sea-ice state: Assuming that the sea-ice initial states are identical for all ensemble members, even slightly different winds will initially tend to drive motion in the same direction because the motion is strongly constrained by the pattern of fractures. Only after some time will the pattern of fractures differ between the ensemble members, and then the sea-ice motion fields will also be more different between the members. Could this not explain why the anisotropy is even larger at the beginning in neXtSIM and then goes down to lower values? This argument of course requires that the initial sea-ice states are identical, so that should be clarified.

P16L11-13: "We found that the ensemble spread follow two distinct diffusion regimes, one for small time t«Gamma and one for large time t»Gamm where Gamma is the so-called integral time scale (Taylor, 1921), which is about 1.5 days for sea ice according to Rampal et al. (2009)"; Do I understand correctly that this integral timescale is quite directly determined by the autocorrelation timescale of wind anomalies or - in the present study - by the autocorrelation timescale of the wind perturbations? It might be worthwile pointung out that this subtle difference exists between the present and the Rampal et al. 2009 study.

[Figure]

P17L1-2: "Predictive skills" and "able to forecast real trajectories"; please see my general comment on the way the term "forecast" is used in this study.

P15L12-14: "We observe that highest degree of ensemble anisotropy (R > 1) is found north of Greenland and Canadian Archipelago, where the ice is the thickest and the ice drift and winds the lowest, in overall agreement with the interpretation of the temporal evolution of R for neXtSIM in the winter"; There are also high values of R along the Eurasian and Alaskan coasts; can't this be explained by the fact that the sea-ice motion (and the associated dispersion) occurs mainly in parallel to the coasts because motion towards the coast tends to be suppressed by counteracting ice pressure (even when the thickness is moderate)?

Fig11: For my taste it would again be better to use the same scale for all panels.

Fig12: If I understand correctly, the slopes at lower timescales are all approximately 2. I suggest to note that also in the plot (as is done for the longer timescales).

P19L10-11: "For FD, $e_L$ still being positive for both periods, corresponds to a drift too far to the right in the observations"; What is meant by "to the right in the observations"? And is $e_L$ for FD not NEGATIVE according to Fig14 right?

P21L10-13: "even if the forecast errors are smaller in neXtSIM than in FD, its shrunk search areas lead to a smaller POC for neXtSIM than for the FD model (not shown): in practice the probabilistic forecast from neXtSIM is too optimistic, underestimates the uncertainties in the forecast, while the FD forecast overestimates them"; First, I would in fact like to see a graph that shows how the spread ($\mu_b$) versus the error evolves. In weather forecasting, the "spread-error relationship" is a common way to measure whether probabilistic forecasts are underdisperive ("too optimistic") or overdispersive ("too pessimistic"). The latter terms could be introduced also in the context of this study.

P22L31-33: "The fact that most of the superiority of neXtSIM over reveals during winter

is, as stated in previous instances, in full agreement with the expectations, given that during the summer the ice mechanics in the two models is similar"; Please check the grammar of this sentence.

Fig17: Could the superiority of FD at very short lead times and for large search areas (for which the skill of the barycenter is not important) be explained by the possibly too strong anisotropy of neXtSIM close to the initial time, due to the shared fracture pattern in all ensemble members (if the sea-ice initial states are indentical, see my previous remarks)?

P24L7-8: "This mechanism is missing in the absence of rheology (like in the FD model) and represents a clear strength and advantage of the elasto-brittle rheology in neXtSIM"; Could the authors comment on what differences one might expect for other rheologies like the standard (E)VP?

P24L18: "The model sensitivity to winds has been evaluated"; Wouldn't it be more precise to say "The model sensitivity to wind perturbations has been evaluated"?

P24L20-P25L1: "we are confident that the spread simulated by the model is physically consistent. Alternative sources of biases must be called such as, for example, other model inputs (thickness, concentrations, damage, ocean currents)"; Deficiencies to simulate reliable spread are commonly not referred to as "biases". Also, what does "must be called" mean here? Maybe in the sense of "must be mentioned?" And why to you refer to those model variables as "inputs"?

**Technical corrections / comments**

P1L6: "10-days" -> "10 days"

P1L10: "in Arctic" -> "in the Arctic"

P1L12: "to of free-drift model" -> "to the free-drift model"

P2L33-34: "Without aiming to make it a key objective."; In terms of grammar, this

seems to be an incomplete sentence.

P3L7: "measures" -> "measurements"

P3L8-13: Please check these lines for grammar (including commas).

P3L19: "stands on the fact"; sounds strange.

P3L31: "as follow" -> "as follows"

P4L3: "Generalities"; I do not think that this term is commonly used this way.

P4L4: "description neXtSIM" -> "description of neXtSIM"

P6L10: "a initial position" -> "an initial position"

P6L22: "explicit mention on the dependence" - "explicit mention of the dependence"

P6L25: "informations" -> "information"

P6L28: "Let consider" -> "Let us consider"

P8L14: "the and the wind" -> "and the wind"

P8L20-21: "ASR reanalysis" -> "ASR" (two times)

P15L3: "On another hand" -> "On the other hand"

P17L11: "average module"; ?

P20L12: "all simulated ensemble of buoys" -> "all simulated ensemble members"

P21L19: "can posed" -> "can be posed"

P22L4: "models comparison" -> "model comparison"

P22L5: "allow as also" -> "allow us also"

P22L27: "larger of about" -> "larger by about"

P23L7: "hold for" -> "hold also for"

P25L18: "a elasto-brittle" -> "an elasto-brittle"

P26L4: "founded" -> "funded"

---

## Referee Comment (RC3) · Anonymous Referee #3 · 23 Nov 2017

**Review of "Probabilistic forecast using a Lagrangian sea ice model: application for search and rescue operations" by Matthias Rabatel, Pierre Rampal, Alberto Carrassi, Laurent Bertino, and Christopher K. R. T. Jones**

**General comments**

The manuscript "Probabilistic forecast using a Lagrangian sea ice model: application for search and rescue operations" by M. Rabatel, P. Rampal, A. Carrassi, L. Bertino, and C.K.R.T. Jones provides a comprehensive evaluation of sea ice drift response to uncertainties in wind forcing using the sea ice model NeXtSIM with elasto-brittle rheology. The authors demonstrate through comparison with what is referred to as a free-drift model anisotropic behavior associated with sea ice mechanical properties in winter, with implications for predictive skill. This paper presents novel concepts and tools to highlight the importance of characterizing sea ice mechanics and rheology for such applications as search and rescue operations in winter. It is recommended that this manuscript be accepted for publication, following consideration of aspects including systematic error in NeXtSIM as documented in earlier studies of this Lagrangian sea ice model, spatial variability in the air drag coefficient, boundary condition sensitivity studies, and further investigation of reasons for discrepancies in dynamics for modeled and observed trajectories. Please find below more specific comments for consideration.

This is also to express agreement with the comments of both reviewers on the quality of manuscript, in addition to statements in regards to justification for term selection in the free drift model, and the need for further description as to how the forecasts are initialized.

**Specific comments**

*Introduction*

p. 2, line 28. In Rampal et al. (2016b), the authors show systematic errors based on comparison of simulated ice drift with the GlobICE dataset (Figure 7). Perhaps note in the Introduction, and provide a figure depicting, the spatial distribution of systematic errors for given timeframes in winter and summer, to distinguish from differences due to compactness and rheology based on comparisons between NeXStSIM and the free drift model. Highlight systematic errors based on comparison with OSISAF.

p. 3, lines 22 – 29. What parameter values are used in the present study, and in particular for compactness (i.e. as in Table 2 in Rampal et al., 2016b)? In the sensitivity analyses for the compactness parameter in Bouillon and Rampal (2015a) it is noted that the opening and closing rates are influenced by the compactness parameter. How are the current wind sensitivity results influenced by the choice of the compactness parameter?

*Sensitivity analysis*

Air drag coefficient and other parameters: Will there be regional variations in the drag coefficients? How is spatial variability in the drag coefficients addressed? Is the calibration method used the same as that in Rampal et al. (2016b)? As previously noted, what value is used for the compactness parameter in this study?

> Specifically:
> p. 9, line 2 and reference to the OSISAF dataset. Are similar results and values obtained for the air drag coefficient using the globeICE drift product for comparison, as in Rampal et al., 2016b?
>
> p. 9, line 6 and p. 8, Figure 2. Is concentration considered to account for spatial variability in the air drag coefficient, as described in Steiner (2001)? In addition, what impact does the drag coefficient have on results?

p. 8, line 15. Perhaps provide justification for this wind speed variance selection (i.e. a value that is 6 times smaller than that used in Sakov et al. (2012).

p. 9, Figure 3. Is it possible to also identify and show systematic errors spatially in another panel in this or a separate figure? Please see previous comments for the Introduction.

p. 10, line 5. 100 km initial spacing. Are results and differences between the NeXtSIM and FD models influenced by different initial spacings?

*Results*

p. 12, Figure 5. Should the contours for the lower panels be the same (i.e. <= 3 for both)? If not, perhaps emphasize the difference in diffusive spread spatial scales for the FD and NeXtSIM models since this, in addition to similarity in spatial patterns between minimum and maximum diffusive spread for both models is of interest and relevant to the present study.

p. 13, Figure 7. Similarly, the contour range should be the same. Sea ice dynamics are different for neXtSIM and FD even in summer. Perhaps include in the text a possible explanation for these differences (i.e. systematic error, parameter selection, FD characterization).

p. 14, line 15. '…effective elastic stiffness E depends non linearly on the ice concentration…' Should this nonlinearity (and spatial variability) also be considered when optimising for the air drag coefficient? Should this too be considered with optimising for the air drag coefficient? Please see previous comments.

p. 14, line 24. 'Where both (winds and ice thickness) are large, \gamma is large'. However, \gamma is also large in the southern Beaufort Sea for large winds and lower ice thickness in winter. Figures depicting maps of \gamma for the NeXtSIM and FD models in winter and summer would also highlight the impacts of ice rheology.

p. 15, Figure 9 caption. 'The PDFs for FD are similar for summer and winter…' Perhaps still show both PDFs in a separate panel with a different y-axis scale.

p. 15, lines 4 – 6. How are lateral boundary conditions (i.e. landfast ice and its extent) addressed in the model? Would sensitivity analyses associated with boundary conditions highlight regional differences in anisotropy and preferential orientation?

p .16 and Figure 12. What are the possible reasons for discrepancies between the observed and modeled ice drift dispersion characteristics and temporal scaling exponents, namely the superdiffusive regime, in summer? Could superdiffusive behavior be attributed to other sources of uncertainty responsible for systematic error in the model?

p. 17, Figure 11. Contour range should be comparable for the FD and NeXtSIM models. Is it possible to use the anisotropy ratio featured in Figure 11 to improve predictive skill for NeXtSIM?

p. 17, line 10. The forecast error vector components should be depicted accurately in Figure 15.

p. 19, Figure 14. How are **e**, **b**, and **a** related when considering the anisotropy ratio and is relation to forecast error? Variance in parallel and perpendicular components of b could also be compared with those for the forecast error in this figure or in figure 12 to demonstrate the anisotropic effects associated with elasto-brittle rheology.

p. 21, line 19. 'for an equal area that can be searched' Does this imply for a fixed area?

p. 25, lines 5 – 7. Would it be possible to quantify these contributions in additional sensitivity analyses?

**Technical corrections**

p. 1, line 12. Replace 'of free-drift' with 'the free-drift'.

p. 2, lines 33 – 34. Combine the sentence 'Without...' with the next sentence.

p. 4, line 10. Change 'spatial' to 'spatially'.

p. 5, line 24. Change 'analysis' to 'analyses'.

p. 6, line 25. Change 'informations' to 'information'

p. 7, Figure 1 figure caption. Perhaps replace 'bouquet' with '-member ensemble'.

p. 11, line 25. Please change to 'Chukchi'

p. 11, line 30. Please replace 'inn' with 'in'

p. 14, line 14 'influences'

p. 19, line 4 Replace 'get very' with 'are'

p. 21, line 19. Insert 'be' prior to 'posed'

p. 22, line 5. Perhaps replace 'allow as also' with 'also allows'

p. 22, line 27. Replace 'of' with 'by'

p. 22, line 30, Perhaps remove 'up'

p. 22, line 31, Perhaps replace 'reveals' with 'FD is observed'

p. 23, line 3, Replace 'sensitivity' with 'sensitive'

p. 23, line 6, Replace 'contrarily' with 'in contrast'

p. 24, line 21, Replace 'called' with 'considered'

p. 25, line 6, Remove 'yet'

**Reference**

Steiner, N., 2001: Introduction of variable drag coefficients into sea ice models, Annals of Glaciology, 33, 181 – 186.

---

## Author Comment (AC1) · 5 Jan 2018

Review of 'Probabilistic forecast using a Lagrangian sea ice model: application for search and rescue operations' by Rabatel et al.

**J.-F. Lemieux (Referee)**

jean-francois.lemieux@canada.ca

First of all, we would like to thank the referee for his in-depth review of the manuscript and his numerous and relevant comments and suggestions. Please find below the answers in blue text to each of the points raised.

NOTE: *In the revised manuscript, we added few words about how we proceeded to optimise the air drag coefficient for the free-drift model, and indicated which value we found. We also updated all the figures showing the results of the new FD simulation and changed the text when describing the results accordingly. Note that it does not change the conclusions of the paper, but modify quantitatively the results we obtain, especially making FD and neXtSIM more similar in the summer.*

title suggestion: Probabilistic forecast using a Lagrangian sea ice model: impact of rheology
*We changed the title following your suggestion to:*
*"Impact of rheology on probabilistic forecast of sea ice trajectories: application for search and rescue operations in the Arctic"*

1. Major comments

1) You need to give more details on how you initialize the forecasts. Do you use fields (h, hs, A, d, u) from the previous forecast? And how do you deal with the FD model? I guess you use the thickness field from neXtSIM as the thickness field from a model without rheology would be completely unrealistic. Please relate that to the caption in Fig. 6.

*Thanks to your comment. Yes indeed, we completely missed to provide explanations on the initial conditions. We use fields from previous neXtSIM simulations on the same period with the same external forcings but without wind perturbations. The text has been updated accordingly (p.11 l.20 and p.12 l.1-2).*

2) I understand why you neglect the rheology term for your FD model. However,

what is the justification for neglecting the inertial term?

*When the rheology is not taken into account, the time scale of the ice dynamics is short (few hours) and the steady state solution (acceleration set to 0) is rapidly reached (see McPhee1980 and Lepparanta2005). Note also that for this study we wanted to use the simplest model as possible, for which an analytical solution can be easily calculated. This is also consistent with Grumbine 1998.*

3) Fig 2. and p. 9 line 3: How do you define FD 'events'? Concentration threshold?

*No concentration threshold here. A FD event is defined as the one when the simulated ice velocity is within a range of 10% around the value of the free-drift solution. By doing so, we avoid using any concentration threshold that can be somehow misleading or at least a poor constraint for defining region of free-drift. The text has been updated accordingly (p.10 l.12-15).*

You have optimized $C_a$ for neXtSIM. Your conclusion (p. 23) says you have done the same thing for the FD model. This should be mentioned and clarified earlier. I also suggest you give the $C_a$ value you obtained for the FD model.

*Thanks to your comment, we realised that the way we optimised the drag coefficient for the free-drift model was not optimal because it used the same geographical restriction as for the neXtSIM model. We therefore re-optimised that parameter for FD, without applying any geographical restriction. Then we re-ran the free-drift simulations for both winter and summer with the newly optimised drag coefficient.*

*In the revised manuscript, we added a short explanation on how we optimised the drag for the free-drift model, and reported on the optimal value we found. We also updated all the figures relative the FD simulation and changed the text describing the results accordingly. Note that, the use of the new (optimised) drag coefficient does not lead to changes in the conclusions of the paper, yet it modifies quantitatively the results we obtain, especially for the summer.*

4) You often discuss spatial correlations between certain fields (e.g. Fig 7 and 8). You relate these high correlations to the rheology and the thickness field. I think it would be a good idea to show maps (winter and summer) of the effective elastic stiffness as it is more representative of the 'strength' of the ice cover than just the thickness field.

*We choose to show the map of ice thickness because, after the concentration, this is the quantity that correlates the most with the drift response to external forcing, and has the advantage of being a meaningful physical variable to everyone. In winter for instance, the concentration is close to 1 everywhere and it is therefore unable to display any spatial correlation (if any). We agree with you that the elastic stiffness is more representative of the "strength" of the ice cover, in a mechanical sense. Nevertheless, except locally (along the LKFs) where the ice is highly damaged, the geographical pattern of elastic stiffness is at first order correlated to the thickness pattern, and have thus opted to show the latter in view of its clearer physical meaning.*

I am also wondering what is the effect of the pressure term? My impression is that the effective elastic stiffness gets very small in summer because the ice is so damaged and cannot heal so that the pressure term plays an important role.

*The pressure term is large only when the local deformation is convergent and if the concentration is close to 100% (See equations 17 and 18 in Rampal et al. 2016b). In summer, it is correct to say that there is no damage healing, and therefore damaged sea ice can only loose mechanical strength over time.*

5) In the comparison of the predictive skill of the model with and without rheology, you look at the error of the barycenter. I think you could also discuss whether the error e(t) of the barycenter is smaller than the one of a single deterministic forecast (no perturbation to the wind). Even if it is not the case, the probabilistic forecasts with its spread would still give important information...

*Indeed, this is a good suggestion. The comparison with the deterministic forecast does provide important information. We have re-run our model to get the deterministic "forecasts", using the ASR reanalysis as forcing and the air drag coefficient equal to 0.0065 (after optimisation with OSI-SAF*

*dataset). We find the forecast error from a single deterministic forecast is close to the probabilistic barycenter, especially in winter, but is larger by 15% in the summer, so there is a small benefit of using the ensemble mean of a probabilistic forecast. Looking at the barycentric coordinates (e_para and e_perp), we note larger differences in the parallel components, whereas the perpendicular components are similar (see Fig. 15 in the revised manuscript).*

I would also be curious about the following experiment...what happens if you move your virtual buoys with the persisted initial velocity of the observed buoys (see Hebert et al. 2015). At what lead time is the ensemble of neXtSIM better than the persisted observed initial velocity? I guess this could give you some indications about the quality of your forcing field.

*While we agree with the Reviewer that the comparison with a persistent forecast could be an interesting experiment to perform, we consider it slightly out of the scope of this work that is centred on the impact of rheology. We hope to be able to investigate this further in the future.*

2. Minor comments

1) Overall the english and the text is very good. There are a few typos. Here is a list of some of them: p.1 line14, p.4 line4, p.11 line29, p.20 line5, p.22 line29, p.25 line28.

*Done*

2) p.2 line 5: Add 'sea ice' before 'forecasting systems'.

*Done*

3) p.2 line 5: Note that RIPS is no longer in operations and has been replaced by the coupled Regional Ice Ocean Prediction System (RIOPS). It would be better to rephrase. The references for this new system are Lemieux et al., 2016 (the paper you already cite) and Dupont et al. 2015: A high-resolution ocean and sea-ice modelling system for the Arctic and North Atlantic oceans.

*Thank you for your note. We updated the name of the prediction system to RIOPS, and we added Dupont et al 2015 as a reference.*

4) p.3 line 3: remove 'advanced'...Just say what it is.

*Done*

5) p.3 line 9: Coon et al., 1974 modeled sea ice as an elasto-plastic material...please rephrase.

*Done*

6) p.3 line 18: 'sea ice responds in a linear way' is vague. Please clarify what you mean by that.

*Done*

7) p.3 line 20: You could add '(due to the limited number of observations)' at the end of this sentence.

*Done*

8) p.3 line 27: Change 'full complexity of the present version' by 'the latest model developments' .

*Done*

9) p.4 line 9: Change 'spatial' by 'spatially'.

*Done*

10) p.4 line 20: Change 'refreezing' by 'freezing'.

*Done*

11) p.4 line 23: What do you mean by 'effective'? Grid cell mean values?.

*Yes, you are correct. The unit of the effective thickness is a volume per unit of area.*

12) p.4 eq. 2: I am not familiar with this formulation of the vector product for the Coriolis term...Don't you want to use the common formulation with the 'x'?

*Yes indeed, we changed the notation.*

13) p.6 line 8: Add 'virtual' before 'buoy'.

*Done*

14) p.6 line 20: I think you need to divide by N in the equation for B(t).

*You are correct. Thank you for having spotted this typo which is now corrected.*

15) The second figure you refer to is Fig.4 (p. 8 line 4). Please change the order.

*Done*

16) p. 8 line 5: Are these 10 m winds? Please specify this and mention the turning angle you use (maybe also for the ocean currents).

*We use turning angles of 0 and 25 degrees for the air and water drags, respectively. These values are now listed in a table that we added in the revised version of the manuscript and in which all the parameters are reported with their respective values. Note also that these values are the same as those used in Rampal et al. 2016b.*

17) p. 9 line 3: Remove 'state-of-art'...Just say what it is.

*Done*

18) p. 9 line 17: 'Dominant' is a bit confusing here because it sounds like it is the largest term in the momentum equation (the wind stress is usually the largest one). Please rephrase.

*Yes, you are right. The word "dominant" is not the right one. We now have rephrased the sentence (p.11 l.11).*

19) p. 10 line 3-5: Why are thermodynamics an issue? You have a thermo model, right?

*There must be a misunderstanding here. The thermodynamics is indeed not an issue. We indeed use a zero-layer thermodynamics model (Semtner, 1976) to melt or form new ice, as well as for the damage healing.*

*What we intended to explain therein was that, in order to ensure a fairer comparison between the free drift model and a model with sea ice rheology, one need to make sure that the sea ice thickness/concentration fields during the simulation is as realistic as possible. This is not the case with the free drift model that, by definition, does not limit the amount of ridging for instance, leading to unrealistic thickening and consequent degrade of the forecast performance in terms of drift. We modified the text accordingly (p11. l.16-20).*

20) p. 10 last line: Add 'steady state' before 'drift'.

*Done*

21) Fig.5: It is difficult to see the coherency between the neXtSIM panels because the lower panel is almost only blue. Can you improve the colorscale so that we can see better the difference? (same idea for Fig. 7)

*We choose this colorscale in order to highlight the quantitative differences between mu_b in neXtSIM and FD. Indeed, in this case, the neXtSIM panels become very blue, especially in winter. We changed the colorscale as suggested in order to highlight the pattern.*

22) p. 11 line 25-30: You mention correlations between spatial fields. Is it just by looking at the figures or you actually calculated spatial correlations?

*This is only a qualitative visual evaluation based on looking at the figures and checking for common patterns. In our case, we considered the correlation looked obvious.*

23) p. 14 line 1: Clarify what you mean by 'the response'...ice velocity? 4

*The response in terms of period averages of mu_r and mu_b. We updated the text as suggested.*

24) p. 15 line 3 and elsewhere: Is 'on another hand' a correct expression? Is it better to use 'on the other hand'?

*Thanks, We did change the text as suggested.*

25) Fig. 13: How do you define the mean sea ice coverage (A=15% contour)?

*Actually, the grey area shows the presence of the sea ice at least during 10 consecutive days (the length of simulations) during the winter or summer period. We added a sentence in the caption figure to clarify this point (Fig.14).*

 Looking at these two panels, as all the buoys are in regions of thick compact ice, it is kind of

obvious that neXtSIM will do better than FD in this experiment. In other words, the FD model would do better if the buoys were uniformly distributed. I would add a sentence to mention that.

*This is correct. We added a sentence as suggested (p.20 l.9-13).*

26) As you calculated the POC for Fig. 17, I suggest you give the exact definition of the POC in eq. 13 instead of saying that it is proportional to...

*This is a good suggestion. We added a sentence with the exact definition of the POC (p.22 l.13-14).*

27) Fig. 17: same idea as before, what happens if you use the deterministic forecast instead of the barycenter? Do you get a real benefit from the ensemble forecast for the time evolution of the POC?

*As explained before: we re-ran deterministic forecasts, using the ASR reanalysis as forcing and the air drag coefficient equal to 0.0065 after optimisation. We obtain similar POCs from ellipses centred on the deterministic forecast (black lines) in winter, whereas they are smaller in summer (see Fig. below).*

[Figure]

28) Please rephrase the last sentence of p. 22.

*Done*

29) p. 23 line 3: replace 'sensitivity' by 'sensitive'.

*Done*

30) p. 25 lines 7-9: I understand what you mean but I find that the two sentences ('Still it is the wind...' and 'we suggest instead...') kind of contradict each other. If the wind is the key player, efforts should be made to improve the forcing winds (by improving the assimilation and forecasts of

the atmospheric model). Just rephrase a bit. By the way I like the discussion about assimilating sea ice fractures...Interesting.

*We agree that the sentence about the wind was confusing and in contradiction with the point we wanted to make. We therefore decided to remove that sentence.*

---

## Author Comment (AC2) · 5 Jan 2018

The comment was uploaded in the form of a supplement:
https://www.the-cryosphere-discuss.net/tc-2017-200/tc-2017-200-AC2-supplement.zip

---

## Author Comment (AC4) · 5 Jan 2018

First of all, we would like to thank the referee for his in depth review of the manuscript and his numerous and relevant comments and suggestions. Please find below the answers in blue text to each of the points raised.

NOTE: *In the revised manuscript, we added few words about how we proceeded to optimise the air drag coefficient for the free-drift model, and indicated which value we found. We also updated all the figures showing the results of the new FD simulation and changed the text when describing the results accordingly. Note that it does not change the conclusions of the paper, but modify quantitatively the results we obtain, especially making FD and neXtSIM more similar in the summer.*

1) Concerning terminology, I think it would be worthwhile to clarify that this study is using the term "forecast" not in the sense of forecasting actual future trajectories, where the future evolution of the system is becoming more and more uncertain over the forecast lead time through chaotic error growth, but in a slightly different way where the future evolution of the most chaotic component - the atmospheric forcing - is approximately known. Of course uncertainties are introduced in another way, namely through perturbations of the atmospheric forcing, but still the underlying synoptic evolution is the same in all ensemble members. I'm not trying to say that this is not worthwhile doing; in particular, one can imagine search & rescue applications where one aims to find the current position of a target that got lost 10 days ago, so one could run a "forecast" system like the one used here to "nowcast" the current position using near-real-time atmospheric (re-)analyses. And, obviously, one could also use actual atmospheric forecasts to drive the model, but that is not done in this study, so I recommend to just clarify this.

*Agree, we specify in the introduction that we work in the context of a "hindcast-forecast" with analysed atmospheric forcing, which error uncertainties are emulated as representative of forecast errors. In the context of data assimilation, this procedure is akin to the propagation step of Ensemble Kalman Filter for generating "forecast" error covariance matrices (which should also be called "hindcast-forecast" to be precise).*

2) P2L33: "departing from independent in situ drifting buoys, and compare them with real observations"; Does "in situ drifting buoys" not refer to the "real observations"? Please clarify.

*Thank you for your comment. No indeed, we replaced the not suitable term: "in situ" by "virtual".*

3) P3L28-29: "the impact of some mechanical parameters on the ice deformation can still be considered as valid"; the "some" sounds very vague, could you be more specific?

*Yes indeed. Actually, all mechanical terms are involved. We changed the text (p.3 l.32).*

4) Eq1: If I am not mistaken, this holds only when h and hs are the "effective" (grid-cell averaged) thicknesses, correct? And for A it stops holding for A close to 1, in particular if there is a lot of damage where there can still be considerable convergence despite A=1 (even with the "pressure term"), right? This would deserve some clarification.

*Yes, the h and hs are indeed volumes per unit of area and A is bounded to 1, even when ridging occurs.*

5) Eq5: It may help to mention what value is used for alpha (probably -20 as in Rampal et al. 2016?) so the strongly non-lineas dependence on A becomes obvious.

*Yes indeed, we used the same alpha value, i.e. -20, as in Rampal et al. 2016b. To make sure this is clear to the reader, we decided to add a table listing all the parameters of the model and the values used for our experiments.*

6) P5L24-28: To me this paragraph sounds very vague; could the authors be more specific on what inputs and outputs are considered?

*We removed that paragraph.*

7) Eq8+9: It might be worth noting that the means of "b_i,II" and "b_i,L" are zero and thus omitted in Eq9. It might also be worth

pointing out that mu_b contains basically the same information as "b_i,II" and "b_i,L" (except the directional information); they do not relate to each other like the first and the second momentum of a distribution (which is not stated, but at least I was confused at first).

*Thank you for your comment. Yes indeed, the means of b_i,|| and b_i,L are zero and highlight only directional information. However, as the directions are perpendicular, we use the sample standard deviation of these distances to define the anisotropy. Then, b_i,|| and b_i,L are no longer used in the paper. We rephrased the paragraph to avoid confusion (end of p.8).*

8) Fig2: What data and analysis is this figure based on? And what temporal sampling frequency is used to detect "events", e.g., one day?

*About events, the temporal sampling frequency used is one day. The ice velocities from these events are compared with the observed ice drift from OSISAF dataset. We updated the text accordingly.*

9) P9L17: "the internal stresses in the ice, and the corresponding Grad(sigma h) term in Eq. (2), becomes very large and dominant"; Would it be more precise to say that it almost completely balances the other forces (so that the acceleration (and speed) becomes very small)?

*Thanks for this suggestion. We rephrased the sentence to avoid confusion; the use of the word "dominant" was indeed not a good choice. This is now clarified in the revised manuscript (p11. l.11).*

10) P10L6: "We ran an ensemble of 12 members, each of them forced by the perturbed wind dataset generated as explained above"; If I have not overseen some important detail, there is some information on the experimental setup missing. In particular, how are the sea ice and ocean in the different members initialised? Is there one single "reference run" from which the ensembles are brached off, with all members keeping the same initial sea-ice/ocean state? If so, does the reference run also have perturbations to the winds (and accordingly uses the re-tuned parameters)? Or are there just 12 simulations overall, covering the whole time period, so that the "initial" sea-ice/ocean states are different between the ensemble members? The latter doesn't seem to be the case as you speak of individual simulations in P10L3. Also, P25L5-6 seems to hint that indeed the initial states are identical. In any case I have the impression that the question of whether or not the initial sea-ice states are identical is very important for the interpretation of some of the results (see below), so I think this should be described very clearly.

*Thanks to your comment: we completely missed to provide explanations on initial conditions. Indeed, all members are initialised with the same sea ice state coming from the reference simulation presented in Rampal et al. 2016b on the same period with same external forcings without wind perturbations. We have now added all the details regarding initialisation in the revised manuscript (end of p.11 and beginning of p.12).*

11) P10L9: "8000 virtual buoy trajectories over the winter season"; Is this the number of ENSEMBLES of buoy trajectories? For individual trajectories I would expect a larger number, given the approximate number of initial positions in Fig4 and the number of 10-days periods.

*You are right. It was confusing in the text, we updated the text stating the total number of trajectories (12 times greater).*

12) Eq11: While you can certainly say that the omitted pressure term, as the stress term, belong to the rheology, the omitted tau_b could also be mentioned.

*Yes, this is a good point. We updated the text accordingly (p.12 l.13).*

13) P10L19: "The FD model therefore mimics the drift of a buoy at the surface of the ocean."; I would think that this is not really the case because the drag coefficients would be quite different (in particular on the water side due to turbulent momentum transfer between deeper layers and the surface water surrounding the buoy)?

*Correct, a buoy in the open ocean would experience a different ocean drag. We use the term « mimics » for the sake of the analogy, given that The free drift case is analogous to the pure Ekman drift case in ocean dynamics. We have rephrased the text to avoid the confusion (p.12 l.16).*

14) Fig5+7: i) I do not understand why the sea-ice thickness pattern is so clearly visible in the dispersion strength (mu_b) for the free-drift model where the rheology shouldn't play any role; could the authors comment? ii) I suggest to use the same colour scales for the two bottom panels so that the difference in mu_b becomes even more obvious

*i) This is because the free drift model (eq. 11) takes into account the ice mass via the Coriolis and gravity force*

*terms. The thickness patterns, which are used as initial conditions and which are the same as in the corresponding neXtSIM simulations, are therefore reflected on both the advective and dispersive response of the FD model.*

*ii) We tried to use the same colorscale for FD and neXtSIM but in this case, on one Figure of them (either FD or neXtSIM), one cannot see any pattern anymore. Finally, we choose to leave different colorscales in order to discuss on mu_b pattern (see Fig. below).*

[Figure]

15) P14L1: "In both winter and summer, the response to wind perturbations is overall lower by 35% in neXtSIM than in FD"; Where does this number come from? I would have thought that the difference of mu_b in neXtSim versus FD would quantify "the response to wind perturbations", but those are reduced by 63% and 39% in winter and summer,

respectively (as stated in P14L5-6), so that doesn't fit. Could you please clarify? (Also at the beginning of Sect.5)

*Thank you for your comment. Actually, this number was a global mean over both periods and both distances, but we removed it and changed the sentence.*

*Yes indeed, the distance $b$ provides quantitative information on the response to wind perturbations. In summer, the reduction shows the behaviour of neXtSIM is closer to FD. This may be explained by the fact that, in summer, the ice concentration is reduced leading to a significant decrease of the internal stresses within the ice (p.13 l.23-32).*

16) P14L21-27: Is the assumption correct that the values found for the ratio mu_r/mu_b should scale with the strength of the wind perturbations? If so, this might be worth mentioning.

*It is an interesting assumption. Besides that, we studied this ratio on different ways: looking at the spatial pattern and the time evolution. However, we could not highlight any relevant correlation with wind perturbations and/or physical quantities. This ratio is roughly driven by mu_r and is not directly related to the wind perturbations.*

17) P15L8-9: "This reveals that the ice will first tend to move compactly along the wind direction away from the origin, but it then starts to break and depart from the barycentre"; First, the wind directions felt by the different ensemble members differ instantly after the initialisation, right?

*You are right.*

18) So, moving compactly along the wind direction would imply a slightly different direction for each member from the very beginning. Second, the ice is "broken" (i.e., has fractures) already at initial time, right?

*You are right, the initial damage is taken from the outputs of simulations used in Rampal et al. (2016b) for neXtSIM (not used in FD) and it is the same for all members.*

19) Third, and maybe more importantly, I think that the interpretation of the decreasing anisotropy might depend strongly on the initial sea-ice state: Assuming that the sea-ice initial states are identical for all ensemble members, even slightly different winds will initially tend to drive motion in the same direction because the motion is strongly constrained by the pattern of fractures.

*We agree with you.*

20) Only after some time will the pattern of fractures differ between the ensemble members, and then the sea-ice motion fields will also be more different between the members. Could this not explain why the anisotropy is even larger at the beginning in neXtSIM and then goes down to lower values? This argument of course requires that the initial sea-ice states are identical, so that should be clarified.

*You are right. We changed the text to highlight this valid interpretation (p18).*

21) P16L11-13: "We found that the ensemble spread follow two distinct diffusion regimes, one for small time t«Gamma and one for large time t»Gamm where Gamma is the so-called integral time scale (Taylor, 1921), which is about 1.5 days for sea ice according to Rampal et al. (2009)"; Do I understand correctly that this integral timescale is quite directly determined by the autocorrelation timescale of wind anomalies or - in the present study - by the autocorrelation timescale of the wind perturbations? It might be worthwhile pointing out that this subtle difference exists between the present and the Rampal et al. 2009 study.

*This is not true. The integral timescale can be influenced by the unperturbed winds and ocean forcing fields, the perturbations autocorrelation and (in the case of neXtSIM), the rheological model. Since the perturbations of the winds have an autocorrelation of 2 days, quite close to the limit of 1.5 days, it is unfortunately not easy to tell their effect apart from other effects, but we have not tried to add additional experiments for that matter.*

22) P17L1-2: "Predictive skills" and "able to forecast real trajectories"; please see my general comment on the way the term "forecast" is used in this study.

*We reminded the reader that this study is "in hindcast mode".*

23) P15L12-14: "We observe that highest degree of ensemble anisotropy (R > 1) is found north of Greenland and Canadian Archipelago, where the ice is the thickest and the ice drift and winds the lowest, in overall agreement with the interpretation of the temporal evolution of R for neXtSIM in the winter"; There are also high values of R along the Eurasian and Alaskan coasts; can't this be explained by the fact that the sea-ice motion (and the associated dispersion) occurs mainly in parallel to the coasts because motion towards the coast tends to be suppressed by counteracting ice pressure (even when the thickness is moderate)?

*Thank you for your comment; this is an interesting point. We agree with you and we updated the text accordingly (p18).*

24) Fig11: For my taste it would again be better to use the same scale for all panels.

*Unfortunately, with the same colorscale, either pattern of neXtSIM or FD will not be longer visible. We choose to keep different colorscale in order to exhibit the absence of spatial coherence for FD on the one hand, and on the other, the difference between ice coverage close to the coast and in the center of the domain.*

25) Fig12: If I understand correctly, the slopes at lower timescales are all approximately 2. I suggest to note that also in the plot (as is done for the longer timescales).

*Yes indeed. We updated the figure as you suggested (Fig. 13).*

26) P19L10-11: "For FD, e_L still being positive for both periods, corresponds to a drift too far to the right in the observations"; What is meant by "to the right in the observations"? And is e_L for FD not NEGATIVE according to Fig14 right?

*Yes indeed. This is an unfortunate typo, we corrected. The vector e is directed from the observation to the barycentre. Thus, standing on the observation and looking in the drift direction, a negative e_L corresponds to seeing the barycentre to the right (Fig. 15).*

27) P21L10-13: "even if the forecast errors are smaller in neXtSIM than in FD, its shrunk search areas lead to a smaller POC for neXtSIM than for the FD model (not shown): in practice the probabilistic forecast from neXtSIM is too optimistic, underestimates the uncertainties in the forecast, while the FD forecast overestimates them"; First, I would in fact like to see a graph that shows how the spread (mu_b) versus the error evolves. In weather forecasting, the "spread-error relationship" is a common way to measure whether probabilistic forecasts are underdisperive ("too optimistic") or overdispersive ("too pessimistic"). The latter terms could be introduced also in the context of this study.

*Thank you for your comment. We added a "spread-error relationship" graphic (Fig. 18 on the revised version) as you suggested. Actually, both models are underdisperive, however, where the spread is larger than 10 km, only neXtSIM seems become too pessimistic. We have added comments on this somewhat surprising behaviour (end of p.23).*

28) P22L31-33: "The fact that most of the superiority of neXtSIM over reveals during winter is, as stated in previous instances, in full agreement with the expectations, given that during the summer the ice mechanics in the two models is similar"; Please check the grammar of this sentence.

*Done. Hopefully improved.*

29) Fig17: Could the superiority of FD at very short lead times and for large search areas (for which the skill of the barycenter is not important) be explained by the possibly too strong anisotropy of neXtSIM close to the initial time, due to the shared fracture pattern in all ensemble members (if the sea-ice initial states are indentical, see my previous remarks)?

*Thank you for your comment. This is an interesting point. Indeed, when we define the search area as a circle (=without anisotropy), the POC from neXtSIM, for small time horizons and for large search areas, becomes greater than FD. For other time horizons, smaller areas and in summer, the difference is far less visible (see Fig. below). This means that the ensemble run is "too confident in its anisotropy", which could be improved with a better initialization of the ensemble. We added this note in the revised manuscript.*

[Figure]

30) P24L7-8: "This mechanism is missing in the absence of rheology (like in the FD model) and represents a clear strength and advantage of the elasto-brittle rheology in neXtSIM"; Could the authors comment on what differences one might expect for other rheologies like the standard (E)VP?

*We have noted some anisotropy in the EVP model in a previous experiment with TOPAZ (Bertino et al., 2015), which could unfortunately not be compared to the present results since it was not carried out in similar conditions.*

*Bertino, L., Bergh, J., and Xie, J.: Evaluation of uncertainties by ensemble simulation, Tech. Rep. Tech. Rep. 355, NERSC, ART JIP Deliverable 3.3, Bergen, Norway, 2015.*

31) P24L18: "The model sensitivity to winds has been evaluated"; Wouldn't it be more precise to say "The model sensitivity to wind perturbations has been evaluated"?

*Yes.*

32) P24L20-P25L1: "we are confident that the spread simulated by the model is physically consistent. Alternative sources of biases must be called such as, for example, other model inputs (thickness, concentrations, damage, ocean currents)"; Deficiencies to simulate reliable spread are commonly not referred to as "biases". Also, what does "must be called" mean here? Maybe in the sense of "must be mentioned?" And why to you refer to those model variables as "inputs"?

*We significantly rephrase these sentences in order to clarify our discussions (p.28 l.23-30).*

**Technical corrections / comments**

P1L6: "10-days" -> "10 days"

*Done*

P1L10: "in Arctic" -> "in the Arctic"

*Done*

P1L12: "to of free-drift model" -> "to the free-drift model"

*Done*

P2L33-34: "Without aiming to make it a key objective."; In terms of grammar, this seems to be an incomplete sentence.

*Done*

P3L7: "measures" -> "measurements"

*Done*

P3L8-13: Please check these lines for grammar (including commas).

*Done*

P3L19: "stands on the fact"; sounds strange.

*Done*

P3L31: "as follow" -> "as follows"

*Done*

P4L3: "Generalities"; I do not think that this term is commonly used this way.

*Done*

P4L4: "description neXtSIM" -> "description of neXtSIM"

*Done*

P6L10: "a initial position" -> "an initial position"

*Done*

P6L22: "explicit mention on the dependence" - "explicit mention of the dependence"

*Done*

P6L25: "informations" -> "information"

*Done*

P6L28: "Let consider" -> "Let us consider"

*Done*

P8L14: "the and the wind" -> "and the wind"

*Done*

P8L20-21: "ASR reanalysis" -> "ASR" (two times)

*Done*

P15L3: "On another hand" -> "On the other hand"

*Done*

P17L11: "average module"; ?

*Done*

P20L12: "all simulated ensemble of buoys" -> "all simulated ensemble members"

*Done*

P21L19: "can posed" -> "can be posed"

*Done*

P22L4: "models comparison" -> "model comparison"

*Done*

P22L5: "allow as also" -> "allow us also"

*Done*

P22L27: "larger of about" -> "larger by about"

*Done*

P23L7: "hold for" -> "hold also for"

*Done*

P25L18: "a elasto-brittle" -> "an elasto-brittle"

*Done*

P26L4: "founded" -> "funded"

*Done*

---

## Author Comment (AC5) · 5 Jan 2018

**Review of "Probabilistic forecast using a Lagrangian sea ice model: application for search and rescue operations" by Matthias Rabatel, Pierre Rampal, Alberto Carrassi, Laurent Bertino, and Christopher K. R. T. Jones**

First of all, we would like to thank the referee for his in depth review of the manuscript and his numerous and relevant comments and suggestions. Please find below the answers in blue text to each of the points raised.

NOTE: *In the revised manuscript, we added few words about how we proceeded to optimise the air drag coefficient for the free-drift model, and indicated which value we found. We also updated all the figures showing the results of the new FD simulation and changed the text when describing the results accordingly. Note that it does not change the conclusions of the paper, but modify quantitatively the results we obtain, especially making FD and neXtSIM more similar in the summer.*

**General comments**

The manuscript "Probabilistic forecast using a Lagrangian sea ice model: application for search and rescue operations" by M. Rabatel, P. Rampal, A. Carrassi, L. Bertino, and C.K.R.T. Jones provides a comprehensive evaluation of sea ice drift response to uncertainties in wind forcing using the sea ice model NeXtSIM with elasto-brittle rheology. The authors demonstrate through comparison with what is referred to as a free-drift model anisotropic behavior associated with sea ice mechanical properties in winter, with implications for predictive skill. This paper presents novel concepts and tools to highlight the importance of characterizing sea ice mechanics and rheology for such applications as search and rescue operations in winter. It is recommended that this manuscript be accepted for publication, following consideration of aspects including systematic error in NeXtSIM as documented in earlier studies of this Lagrangian sea ice model, spatial variability in the air drag coefficient, boundary condition sensitivity studies, and further investigation of reasons for discrepancies in dynamics for modeled and observed trajectories. Please find below more specific comments for consideration.

This is also to express agreement with the comments of both reviewers on the quality of manuscript, in addition to statements in regards to justification for term selection in the free drift model, and the need for further description as to how the forecasts are initialized.

**Specific comments**

*Introduction*

1) p. 2, line 28. In Rampal et al. (2016b), the authors show systematic errors based on comparison of simulated ice drift with the GlobICE dataset (Figure 7). Perhaps note in the Introduction, and provide a figure depicting, the spatial distribution of systematic errors for given timeframes in winter and summer, to distinguish from differences due to compactness and rheology based on comparisons between NeXStSIM and the free drift model. Highlight systematic errors based on comparison with OSISAF.

*Thank you for your comment. We have added, in the introduction, a figure depicting the spatial distribution of systematic errors in winter. We do not provide the figure in summer since we do not consider the OSISAF data to be sufficiently reliable in this period (Fig. 1, p.4 l.1-5).*

2) p. 3, lines 22 – 29. What parameter values are used in the present study, and in particular for compactness (i.e. as in Table 2 in Rampal et al., 2016b)? In the sensitivity analyses for the compactness parameter in Bouillon and Rampal (2015a) it is noted that the opening and closing rates are influenced by the compactness parameter. How are the current wind sensitivity results influenced by the choice of the compactness parameter?

*You are right, we used the same compactness as in Rampal et al 2016b. A list of the values of the parameters was missing in our submitted manuscript. We have now added a table listing those in the revised manuscript.*

*In this study, we decided to restrict the sensitivity analysis to external parameters only, here the wind, and not to extent it to the internal mechanical parameters of sea ice like compactness, cohesion, etc. This choice is further justified by our mid-term goal of using neXtSIM in conjunction with ensemble-based data assimilation in which context the ensemble would preferably reflect the uncertainty on the external forcing under the assumption that internal parameters have all been already optimised.*

*We agree however with Reviewer on the relevance of such an analysis but we believe that it is beyond the scope of this paper, and it will be addressed in a different study.*

*Sensitivity analysis*

3) Air drag coefficient and other parameters: Will there be regional variations in the drag coefficients?

*There are no regional variations in this present study, both for the sake of simplicity and because constant drag coefficients are still customary in the community.*

4) How is spatial variability in the drag coefficients addressed?

*This is not addressed in the present study. We assume this coefficient to be constant over time and over the Arctic*

5) Is the calibration method used the same as that in Rampal et al. (2016b)?

*Yes, indeed. This is the same method used here. We explain it better in the revised manuscript, and also specify the values we obtain for neXtSIM and FD.*

6) As previously noted, what value is used for the compactness parameter in this study?

*We use the same one as in Rampal et al 2016b, i.e. -20.*

7) Specifically: p. 9, line 2 and reference to the OSISAF dataset. Are similar results and values obtained for the air drag coefficient using the globeICE drift product for comparison, as in Rampal et al., 2016b?

*We have not done any comparison to GlobICE in this paper as it is important for this specific study (and its goals) to perform the optimisation of the drag over the whole arctic, which would not have been the case if using GlobICE that has significantly less spatial coverage.*

8) p. 9, line 6 and p. 8, Figure 2. Is concentration considered to account for spatial variability in the air drag coefficient, as described in Steiner (2001)?

*As said before, we do not consider any spatial variability in the drag coefficient in this study. The concentration is actually not directly used to account for any spatial variability of the drag coefficient. But indirectly it is so, since we perform the optimisation only where the simulated drift is close to the free-drift solution, which*

*happens to be at locations where the concentration is significantly lower than 100%.*

9) In addition, what impact does the drag coefficient have on results?

*Although this is an interesting question, this paper is not intended to address it. Still, between the submission and the present review, the drag coefficient of the FD model has been reduced from 5.1 e-03 down to 3.2 e-03, which has changed quantitatively the results in the summer but did not invalidate the conclusions.*

10) p. 8, line 15. Perhaps provide justification for this wind speed variance selection (i.e. a value that is 6 times smaller than that used in Sakov et al. (2012).

*Note that the value of 6 may sound dramatic while it only makes a factor of 2.3 in standard deviation. If taking the variance used in Sakov et al 2012, the impact on the neXtSIM behaviour is too large, i.e. the ice is breaking up too much leading to excesses of ice drift and very small anisotropy of the ensemble. We therefore decided to reduce that variance to a reasonable level so that the physics of neXtSIM can be expressed. In the future we will compare the relative sensitivities of the model used in Sakov et al. (2012) and neXtSIM.*

11) p. 9, Figure 3. Is it possible to also identify and show systematic errors spatially in another panel in this or a separate figure? Please see previous comments for the Introduction.

*See comment 1*

12) p. 10, line 5. 100 km initial spacing. Are results and differences between the NeXtSIM and FD models influenced by different initial spacings?

*We have not tested this.*

*However, if using e.g. 50km, we may not be able to consider anymore that a given drift trajectory of 10days is independent of each other as a given virtual buoy would sample more than one "box" over that period (average sea ice speed is about 6km/day in winter). So, taking 100km almost ensure that the trajectory members started from the centre of one box are independent of the trajectory members of the neighbouring box.*

*Results*

13) p. 12, Figure 5. Should the contours for the lower panels be the same (i.e. <= 3 for both)? If not, perhaps emphasize the difference in diffusive spread spatial scales for the FD and NeXtSIM models since this, in addition to similarity in spatial patterns between minimum and maximum diffusive spread for both models is of interest and relevant to the present study.

*We adjusted the colorscale as you suggested (Fig. 6).*

14) p. 13, Figure 7. Similarly, the contour range should be the same. Sea ice dynamics are different for neXtSIM and FD even in summer. Perhaps include in the text a possible explanation for these differences (i.e. systematic error, parameter selection, FD characterization).

*We tried to use the same color scale for FD and neXtSIM but in this case, on one Figure of them (either FD or neXtSIM), one cannot see any pattern anymore. Finally, we choose to leave different colorscales in order to discuss on mu_b pattern.*

15) p. 14, line 15. '...effective elastic stiffness E depends non linearly on the ice concentration...'

Should this nonlinearity (and spatial variability) also be considered when optimising for the air drag coefficient? Should this too be considered with optimising for the air drag coefficient? Please see previous comments.

*Such an optimisation of the drag where the rheology is active represents a complex non-linear inverse problem, highly sensitive to poorly known initial values (the ice damage and ice thickness among others). Our optimisation using free drift "events" is precisely intended to solve a simpler linear problem still using a sufficiently large number of observations.*

16) p. 14, line 24. 'Where both (winds and ice thickness) are large, \gamma is large'. However, \gamma is also large in the southern Beaufort Sea for large winds and lower ice thickness in winter. Figures depicting maps of \gamma for the NeXtSIM and FD models in winter and summer would also highlight the impacts of ice rheology.

*Thank you for your comment. Yes, indeed, both winds and ice thickness are not the only explanation. Perhaps, we may add: the sea-ice motion occurs mainly in parallel to the coasts because motion towards them tends to be suppressed by counteracting ice pressure. In summer, these coasts do no longer play the role of closed boundaries and the increase of $R$ is almost no visible. This is corroborated by observing the pattern from FD where the pressure term does not interfere. We updated the text accordingly (end of p.18).*

17) p. 15, Figure 9 caption. 'The PDFs for FD are similar for summer and winter...' Perhaps still show both PDFs in a separate panel with a different y-axis scale.

*Thank you for your comment. We present this figure below. We believe it is not helpful to add it to the revised manuscript.*

[Figure]

18) p. 15, lines 4 – 6. How are lateral boundary conditions (i.e. landfast ice and its extent)

addressed in the model? Would sensitivity analyses associated with boundary conditions highlight regional differences in anisotropy and preferential orientation?

*We are not sure to well understand the question of the referee. However, here is our answer:*

*The lateral boundary conditions are either "free" (if at the ice edge) of "fixed" (if at the coast). If the ice cover does not extend anymore to the coast, the boundary conditions are therefore very different, and this likely has an impact at least over the peripheral band of ice near the ice edge. Further into the ice pack, the impact of the boundary conditions on the sea ice drift and anisotropy of the dispersion is less important. Local sea ice conditions (compactness/concentration and damage) are in this case more likely to be responsible for the anisotropy of the dispersion, which is what we discuss in our study.*

*As a conclusion, we do not think that performing sensitivity analyses associated with boundary conditions would reveal key information to understand the source of anisotropy of the ensemble spread.*

19) p .16 and Figure 12. What are the possible reasons for discrepancies between the observed and modeled ice drift dispersion characteristics and temporal scaling exponents, namely the superdiffusive regime, in summer? Could superdiffusive behavior be attributed to other sources of uncertainty responsible for systematic error in the model?

*We suggest that the super-diffusive behaviour we obtain for summer 2008 with neXtSIM, and which is in apparent contradictions with the results of Rampal et al. (2009) could rather be the fingerprint of a change in sea ice dynamical regime that occured over the most recent years, as a consequence of the thinner and more mobile sea ice cover . In this case, it would mean that the effect of the rheology became weaker (if not absent) in summer and that the sea ice response is now more directly related to ocean currents and winds, and therefore can exhibit super-diffusive behaviour as also reported in Lukovich et al. (2015). We modified the text accordingly (end of p.19 and beginning of p.20).*

20) p. 17, Figure 11. Contour range should be comparable for the FD and NeXtSIM models. Is it possible to use the anisotropy ratio featured in Figure 11 to improve predictive skill for NeXtSIM?

*Unfortunately, if we use the same colorscale, either one of the patterns of neXtSIM or FD will disappear. We choose to keep different colorscale in order to exhibit the absence of spatial coherence for FD on the one hand, and on the other, the difference between ice close to the coast and ice in the center.*

21) p. 17, line 10. The forecast error vector components should be depicted accurately in Figure 15.

*We updated the Fig. 15 (16 in the revised manuscript) as you suggested.*

22) p. 19, Figure 14. How are **e**, **b**, and **a** related when considering the anisotropy ratio and is relation to forecast error? Variance in parallel and perpendicular components of b could also be compared with those for the forecast error in this figure or in figure 12 to demonstrate the anisotropic effects associated with elasto-brittle rheology.

*The comparison of ensemble spread and errors on the same graph would not be helpful because the spread is underestimated by both models (see the new Figure 18). Also, note the answer to a related question (29)) from Reviewer #2, and the complementary graph below. The strong anisotropy may remain more of a hindrance than an advantage to search forecasting as long as the deformations are not assimilated in the model.*

[Figure]

23) p. 21, line 19. 'for an equal area that can be searched' Does this imply for a fixed area?

*Not fixed in the sense that the geometry is different (circle versus ellipse) the area included in the ellipse/circle is the same. Thus the sentence should be understood as: taking an ellipse (or circle) from either models with the same encompassed area, which of the two is more likely to contain the object? We have replaced equal by "a given area", hoping this will be clearer (p.25 l.5).*

24) p. 25, lines 5 – 7. Would it be possible to quantify these contributions in additional sensitivity analyses?

*The contribution of ice drift to the TOPAZ system with respect to other assimilated observations is quantified in Sakov et al. (2012) using the Degrees of Freedom for Signal. neXtSIM does not assimilate the same observations but the maps of $\mu_b$ in Figure 6 represent a sensitivity analysis of ice drift to spatially and temporally stationary perturbations of the winds.*

**Technical corrections**

p. 1, line 12. Replace 'of free-drift' with 'the free-drift'.

*Done.*

p. 2, lines 33 – 34. Combine the sentence 'Without…' with the next sentence.

*Done*

p. 4, line 10. Change 'spatial' to 'spatially'.

*Done*

p. 5, line 24. Change 'analysis' to 'analyses'.

*Done*

p. 6, line 25. Change 'informations' to 'information'

*Done*

p. 7, Figure 1 figure caption. Perhaps replace 'bouquet' with '-member ensemble'.

*Done*

p. 11, line 25. Please change to 'Chukchi'

*Done*

p. 11, line 30. Please replace 'inn' with 'in'

*Done*

p. 14, line 14 'influences'

*Done*

p. 19, line 4 Replace 'get very' with 'are'

*Done*

p. 21, line 19. Insert 'be' prior to 'posed'

*Done*

p. 22, line 5. Perhaps replace 'allow as also' with 'also allows'

*Done*

p. 22, line 27. Replace 'of' with 'by'

*Done*

p. 22, line 30, Perhaps remove 'up'

*Done*

p. 22, line 31, Perhaps replace 'reveals' with 'FD is observed'

*Done*

p. 23, line 3, Replace 'sensitivity' with 'sensitive'

*We are used to seeing the wording "sensitivity experiment" rather than "sensitive experiment" which we interpret as an experiment dealing with a sensitive topic. So we would prefer keeping "sensitivity" (p.27, l.6).*

p. 23, line 6, Replace 'contrarily' with 'in contrast'

*Done*

p. 24, line 21, Replace 'called' with 'considered'

*Done*

p. 25, line 6, Remove 'yet'

*Done*

**Reference**

Steiner, N., 2001: Introduction of variable drag coefficients into sea ice models, Annals of Glaciology, 33, 181 – 186.